# Targeted gene editing and near-universal cDNA insertion of *CYBA* and *CYBB* as a treatment for chronic granulomatous disease

Jonas Holst Wolff [1], Thomas Wisbech Skov[1], Didde Haslund [2],
Sofie Rahbek Dorset[1], Anne Louise S. Revenfeld[2], Clotilde Aussel[3],
Sofie E. Jørgensen[4], Mette Holm[5,6], Martin K. Thomsen [1], Sandra Ammann [3,7],
Toni Cathomen [3,7], Trine H. Mogensen [1,4], Bjarne Kuno Møller[2],
Rasmus O. Bak [1] & Jacob Giehm Mikkelsen [1] ✉

Chronic granulomatous disease (CGD) is a severe inborn error of immunity caused by NADPH oxidase defects. Here, we develop CRISPR/Cas9-based gene editing strategies for correction of variants in the *CYBA* and *CYBB* genes causing CGD. For X-linked CGD, we also develop a near-universal gene editing strategy by targeted integration of a truncated *CYBB* cDNA in CD34+ hematopoietic stem and progenitor cells (HSPCs). Throughout, off-target editing and chromosomal translocations are evident, which negatively impact the ability of gene-edited HSPCs to engraft in immunodeficient mice. However, by employing a high-fidelity Cas9 to minimize off-target editing, we demonstrate restoration of the multilineage engraftment potential of gene-edited HSPCs. Moreover, to further improve safety, we develop a D10A Cas9n editing approach with no detectable off-target activity or chromosomal translocations. Collectively, through risk assessments of different gene editing approaches, we present a D10A Cas9n-based strategy with improved safety, offering a potentially curative treatment for CGD patients.

Chronic granulomatous disease (CGD) is an inborn error of immunity (IEI) with a prevalence of approximately 1:200,000[1]. It is caused by defects in the multisubunit NADPH oxidase, leading to defective generation of reactive oxygen species (ROS) in phagocytic neutrophil granulocytes[2]. As a result, patients with CGD suffer from life-threatening bacterial and fungal infections due to impaired microbial killing[3]. CGD can arise from loss-of-function variants in the *CYBB* gene encoding the main catalytic subunit of NADPH oxidase, NOX2 (gp91phox), leading to

X-linked CGD (X-CGD), accounting for 70% of CGD cases. Alternatively, CGD can be caused by variants in genes encoding one of four other subunits (p47phox, p22phox, p67phox, and p40phox), leading to autosomally inherited forms of CGD. Of the autosomal forms of CGD, variants in the *CYBA* gene (p22phox) account for 15–30% of cases[2].

The current treatment for CGD relies on allogeneic hematopoietic stem cell (HSC) transplantation, but it remains challenging to identify matched donors, and the treatment entails a high risk of developing

[1]Department of Biomedicine, Aarhus University, Aarhus, Denmark. [2]Department of Clinical Immunology, Aarhus University Hospital, Aarhus, Denmark. [3]Institute for Transfusion Medicine and Gene Therapy, Medical Center – University of Freiburg, Freiburg, Germany. [4]Department of Infectious Diseases, Aarhus University Hospital, Aarhus, Denmark. [5]Department of Paediatrics and Adolescent Medicine, Aarhus University Hospital, Aarhus, Denmark. [6]Department of Clinical Medicine, Aarhus University Hospital, Aarhus, Denmark. [7]Center for Chronic Immunodeficiency, Faculty of Medicine, University of Freiburg, Freiburg, Germany. ✉e-mail: giehm@biomed.au.dk

graft-versus-host disease, graft failure, or infections[4]. Additionally, ex vivo lentiviral gene transfer to autologous CD34+ hematopoietic stem and progenitor cells (HSPCs) is not well suited as a treatment for CGD, as proper physiological gene regulation of NADPH oxidase proteins has been shown to be required for hematopoietic stem cell function[5]. Treatment of CGD patients has thus largely been prophylactic to prevent or treat infections, leading to patients relying on life-long administration of antibiotics or antifungal therapy[2]. Consequently, transplantation of autologous ex vivo gene-edited HSCs has emerged as a promising treatment option for IEIs, including CGD[6–10]. Multiple studies have previously described the use of CRISPR/Cas9[11,12] gene correction in CD34+ HSPCs by homology-directed repair (HDR) as a potential treatment for various blood disorders, including X-linked CGD[10,13], X-linked hyper-IgM syndrome[14,15], severe combined immunodeficiency (SCID)-X1[16,17], Wiskott–Aldrich Syndrome[18], β-thalassemia[19,20], and sickle cell disease (SCD)[6,7,21–28]. Although these studies have shown promising results, they have also highlighted key challenges of CRISPR/Cas9-based gene correction that hinder the clinical translation of these approaches. Such challenges include off-target editing, chromosomal aberrations, and p53 activation, leading to decreased fitness of gene-edited HSPCs and lower engraftment potential[29–33]. Additionally, several xenograft studies have found editing rates to be significantly lower in the long-term repopulating HSCs (LT-HSCs), indicating that these cells are less susceptible to undergo HDR and/or more prone to be negatively impacted by the gene editing procedure[6,7,33].

To harness the cellular HDR machinery for repair of the CRISPR/Cas9-mediated DNA double-strand break (DSB), leading to gene correction, a DNA repair template must be provided to the cells. While both single-stranded oligodeoxynucleotides (ssODNs)[7,10], integrase-defective lentiviral vectors (IDLVs)[34,35], and recombinant adeno-associated virus serotype 6 (rAAV6)[6,17] have been used previously with success as HDR templates in CD34+ HSPCs, most studies have focused on rAAV6 due to the high HDR rates that are generally observed with this vector. However, a recent study found that rAAV6 exacerbated the DNA damage response (DDR) caused by the CRISPR/Cas9-mediated DSBs and advocated for the use of IDLV as an alternative repair template platform[32,34]. Similarly, a recent study found that ssODN-based gene editing of CD34+ HSPCs led to higher CFU potential and engraftment capability, when compared to rAAV6-based gene editing[36].

Here, we develop and optimize CRISPR/Cas9-based gene editing strategies for the correction of two CGD-causing variants identified in patients at Aarhus University Hospital in Denmark; a *CYBA* c.287+1 G > T variant (p22phox) causing autosomal CGD and a *CYBB* c.252 G > A variant (NOX2) causing X-CGD. We compare ssODN, IDLV, and rAAV6 for use as repair templates in CD34+ HSPCs and develop optimized gene editing strategies by including mRNA-encoded inhibitors/effectors previously shown to dampen the p53 response and enhance HDR rates and cell fitness. In addition, using rAAV6, we show efficient insertion of a truncated *CYBB* cDNA (consisting of exon 3 through 13) in CD34+ HSPCs from a heterozygous carrier, leading to functional ROS production. We show successful engraftment of gene-edited CD34+ HSPCs in xenotransplanted mice. Using standard SpCas9 and a high-fidelity SpCas9 variant, we show differences in off-target effects with substantial impact on levels of cytotoxicity and multilineage engraftment potential in mice. To improve safety further, we develop a paired Cas9 nickase gene editing strategy of *CYBB* that does not lead to detectable off-target editing or formation of chromosomal translocations.

## Results

### RNP- and rAAV6-based gene editing of a *CYBA* variant impairs HSPC fitness

We first investigated if we could target the *CYBA* c.287+1 G > T locus effectively by inserting a SFFV-driven GFP cassette into the *CYBA* gene,

using nucleofection of Cas9/sgRNA ribonucleoprotein complexes (RNPs) coupled with template delivery using rAAV6[6] (Fig. 1a). Initially, four different sgRNAs were tested for the potency in generating indels close to the patient variant (Fig. 1b) and using the most potent sgRNA (sg178) with an optimized rAAV6 dose of 5000 vector genomes cell⁻¹ (vg cell⁻¹), we were able to achieve up to 40% targeted integration of the GFP cassette in CD34+ HSPCs from healthy donors (Fig. 1c, Supplementary Fig. 1). However, the genome-edited HSPCs showed notably decreased proliferation following gene editing (Fig. 1d), in line with previous studies reporting significant DNA damage response (DDR) induction in CD34+ HSPCs upon RNP nucleofection and rAAV6 transduction[32,34,37]. Similarly, when RNP + AAV-treated HSPCs were subjected to a colony-forming unit (CFU) assay, RNP + AAV-treatment led to a marked decrease in number of colonies formed (Fig. 1e), although we did not observe any overall difference in the distribution of HSPC-derived colonies between untreated HSPCs and RNP + AAV-treated HSPCs (Supplementary Fig. 2). The percentage of GFP+ colonies was also comparable with the targeted integration levels in bulk HSPCs, indicating that the gene editing did not affect myeloid or erythroid differentiation (Fig. 1f). Furthermore, we investigated the multilineage and long-term repopulation capacity of the RNP + AAV-treated HSPCs by xenotransplantation of irradiated, immunodeficient NOG mice (Supplementary Fig. 3a). In accordance with our findings on CFU potential, we found that RNP + AAV-treated HSPCs had an impaired ability to repopulate the bone marrow of immunodeficient mice (Fig. 1g, h, Supplementary Fig. 3b–d), accompanied by a significant decrease in targeted integration levels at 16 weeks post engraftment (Fig. 1i).

### Optimized gene editing approaches of *CYBA* lead to functional ROS production

While CRISPR/Cas9-induced DSBs have long been known to initiate a DNA damage response and p53 activation, recent studies have shown that rAAV6 independently induces p53 activation in CD34+ HSPCs, affecting both cell fitness as well as engraftment potential of gene-edited HSPCs[32–34], which is in line with results of this study. To combat these issues, multiple strategies have been published to both directly dampen the p53 response and simultaneously increase the HDR rates in CD34+ HSPCs. De Ravin and colleagues reported that transient inhibition of p53-binding protein 1 (53BP1) leads to increased HDR rates in CD34+ HSPCs[10,38]. Additionally, Ferrari and colleagues reported that inhibition of p53 by transient ectopic expression of the dominant negative p53 truncated variant, GSE56, increased the polyclonal engraftment ability of CD34+ HSPCs, while transient expression of the adenovirus 5 E4orf6/7 (Ad5-E4orf6/7) protein increased HDR rates in LT-HSCs[33]. We therefore sought to investigate the effect of these inhibitors/proteins on targeted integration into the *CYBA* gene. We produced in vitro-transcribed mRNA of i53 (53BP1 inhibitor), GSE56 (p53 inhibitor), and Ad5-E4orf6/7 and included these mRNA transcripts in the RNP nucleofection. In agreement with previous studies[33,38], we observed increased HDR rates when using i53 and GSE56 + Ad5-E4orf6/7, with the highest effect observed when combining all three mRNA transcripts, which increased HDR rates from 55% without mRNA transcripts to 61% with all mRNA transcripts (Fig. 2a). The effect on HDR was also evident in the LT-HSC-enriched CD34+CD45RA−CD90+ subset, which also showed significantly higher viabilities and total number of edited cells, indicating overall increased fitness of gene-edited cells (Fig. 2b, Supplementary Fig. 4a–c). We therefore decided to include both i53, GSE56, and Ad5-E4orf6/7 mRNA transcripts in all subsequent experiments.

Next, we set out to test and compare rAAV6, ssODNs, and integrase-deficient lentiviral vectors (IDLVs) for use as repair templates. To do this, we designed an HDR template to correct the *CYBA* c.287+1 G > T variant and simultaneously install silent mutations to disrupt the sg178 binding site (Fig. 2c). The HDR template would also facilitate correction of other known pathogenic variants in the vicinity

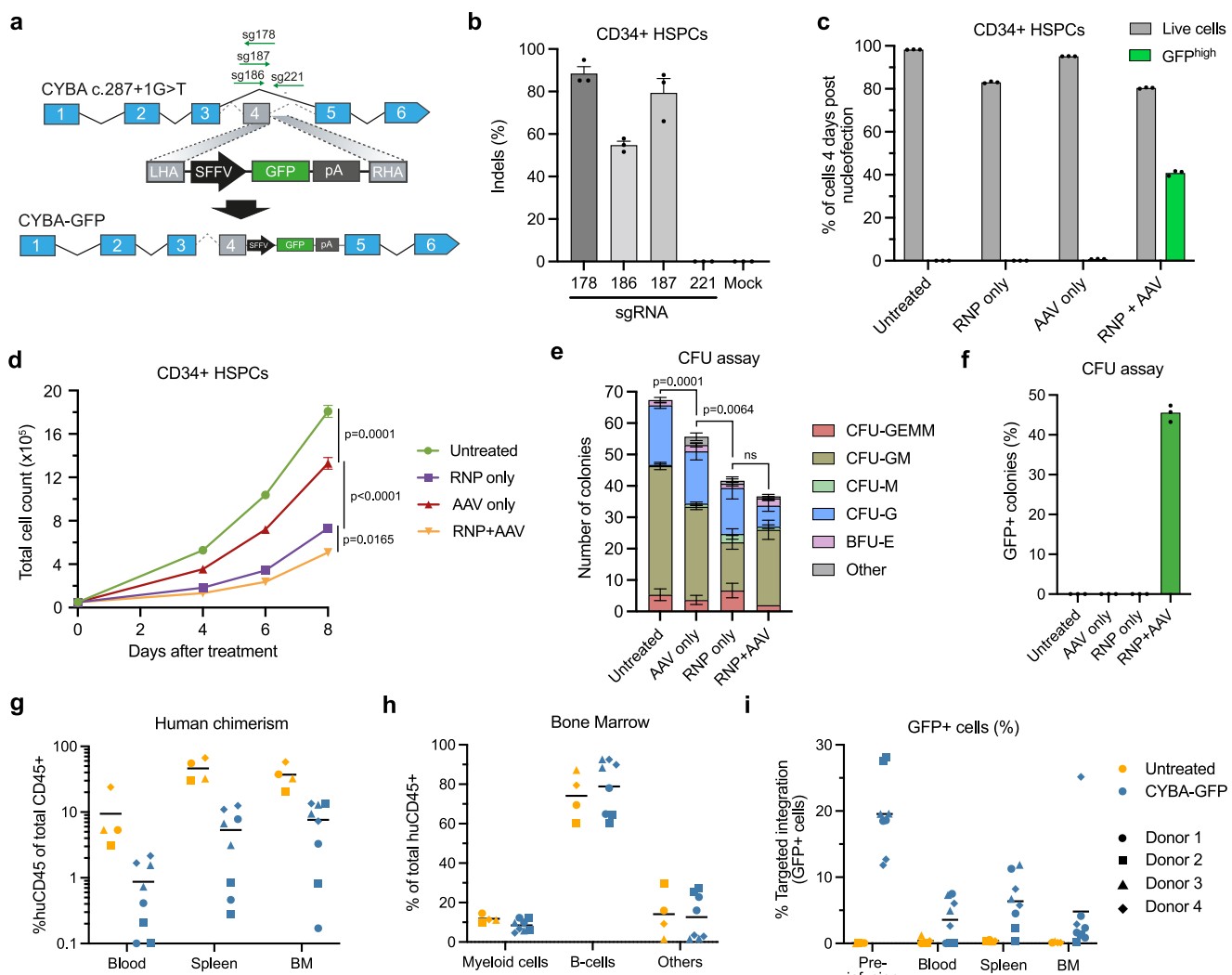

**Fig. 1 | RNP + AAV-based targeted insertion in *CYBA*. a** Schematic representation of the targeted insertion of an SFFV-GFP cassette into the *CYBA* c.287+1 locus. **b** Percentage of indel formation in CD34[+] HSPCs from healthy donors using different sgRNAs targeting sites close to the *CYBA* c.287+1 G > T variant. **c** Targeted integration levels and viabilities of treated HSPCs 4 days after nucleofection. Targeted integration levels were determined as the percentage of GFP[high] cells. **d** Proliferation of treated HSPCs measured by total cell count up to 8 days after nucleofection. **e** Flow cytometry-based CFU assay of treated HSPCs showing total colony counts and distribution of HSPC-derived colonies. **f** The percentage of GFP[+] colonies observed in the CFU assay. **g** Human chimerism in injected mice was measured in the blood, spleen, and bone marrow (BM) 16 weeks after injection. Experiments were conducted using HSPCs from 4 individual healthy donors. **h** The distribution of lineages within the human graft of untreated HSPCs (*n* = 4) and *CYBA* gene-edited HSPCs (*n* = 8). **i** Targeted integration levels in the human graft 16 weeks after injection of untreated HSPCs (*n* = 4) and *CYBA* gene-edited HSPCs (*n* = 8). Data represented as mean (*n* = 3 biological replicates) ± standard deviation unless otherwise specified. Statistical significance was determined by one-way ANOVA with Tukey's multiple comparisons test. For the CFU assay, total colony numbers were used for the ANOVA test. Source data are provided as a Source Data file.

of *CYBA* c.287+1 G > T (Supplementary Fig. 5a), and at the same time, the silent mutations allowed us to determine the degree of gene editing in CD34[+] HSPCs from healthy donors. For rAAV6 vectors, we used a repair template containing homology arms (HA) of 400 bp and a dose of 5000 vg cell[−1] (Supplementary Fig. 5b–d). For ssODNs, we used 200-nt sense ssODNs of which 100 pmol were included in the nucleofection based on previous studies[7] (Supplementary Fig. 5d). For gene editing using IDLV, we used an identical HDR template and transduced HSPCs using a protocol based on previous studies that used two consecutive transductions in the presence of Cyclosporin H (CsH)[34,39]. We also included a protocol that relied on a single transduction using LentiBoost and RetroNectin-coated plates as transduction enhancers (Supplementary Fig. 6). Based on this setup, we found that rAAV6 templates resulted in markedly higher levels of HDR than ssODN and IDLV, even when IDLV transduction was increased with CsH (Fig. 2d). Cell viabilities and proliferation were also similar or better (Fig. 2d).

with rAAV6 relative to ssODNs and IDLVs (Fig. 2e). Therefore, we moved on with rAAV6 as repair template with all three mRNA transcripts included in the RNP nucleofection. Using this optimized HDR protocol, we were able to achieve potent gene editing of the *CYBA* c.287+1 G > T locus in CD34[+] HSPCs from a separate healthy donor, reaching >40% gene editing (Fig. 2f). Additionally, the gene-edited HSPCs showed functional ROS production when differentiated into granulocytes (Fig. 2g, h, Supplementary Fig. 7a, b), reaching up to 60% of that of mock-treated cells. It has been reported that 20% ROS[+] neutrophils in X-CGD patients can be sufficient to provide a level of neutrophil function to support protection from CGD-typical infections, suggesting that the percentage of ROS[+] cells that we measured was well above the therapeutic threshold[40]. Together, these results demonstrated that we could optimize the gene editing strategy to obtain higher HSPC fitness, leading to robust gene editing of the *CYBA* gene and functional ROS production.

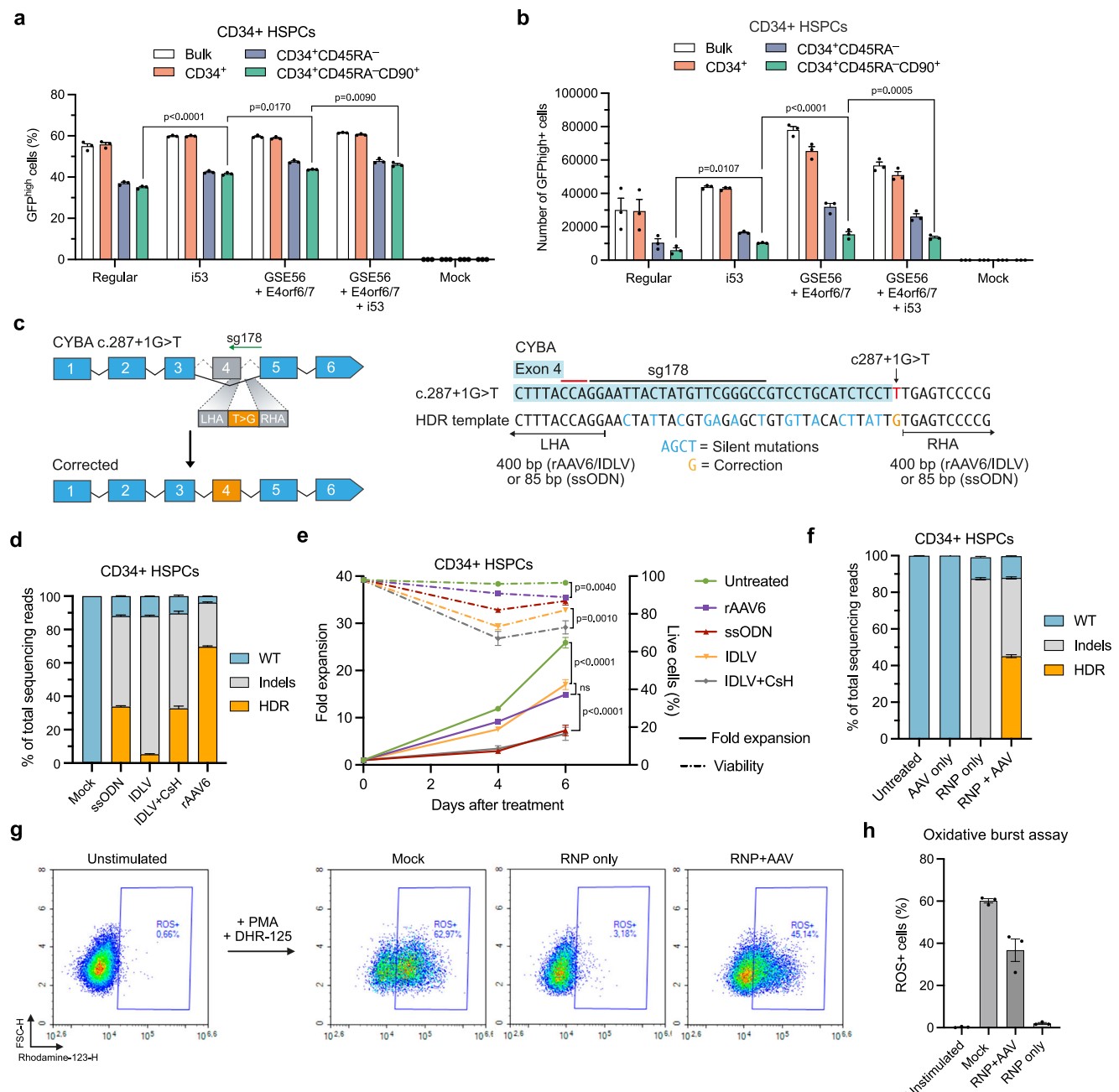

**Fig. 2 | Optimized gene editing approaches of *CYBA*. a** Targeted integration (GFP^high) levels in different HSPC subpopulations using i53, GSE56, and Ad5-E4orf6/7 mRNA transcripts measured by flow cytometry 4 days after nucleofection. **b** Total number of cells with targeted integration (GFP^high). **c** Schematic representation of the gene correction strategy for the *CYBA* c.287+1 G > T variant. The HDR template was designed to install silent mutations at the c.287+1 G > T site to allow quantification of gene editing in CD34^+ HSPCs from healthy donors. **d** Gene editing in healthy donor CD34^+ HSPCs using ssODNs, IDLV, or rAAV6 for delivery of HDR templates. Editing rates were quantified by NGS 4 days after nucleofection. **e** The proliferation and viability of cells from (**d**) up to 6 days after nucleofection. **f** Gene editing in CD34^+ HSPCs from a separate HSPC donor using RNP + AAV. **g** Representative flow cytometry plots of the oxidative burst assay in granulocytes differentiated from treated HSPCs in (**f**). ROS production was measured by the presence of Rhodamine-123. **h** Quantification of oxidative burst assay in differentiated granulocytes. Data represented as mean (*n* = 3 biological replicates) ± standard deviation. Statistical significance was determined by one-way ANOVA with Tukey's multiple comparisons test. Source data are provided as a Source Data file.

## Allele-specific gene editing of X-CGD-causing *CYBB* variant

Although the results so far demonstrate efficient gene editing of the *CYBA* c.287+1 G > T variant at therapeutically relevant levels, we were not able to acquire patient-derived CD34^+ HSPCs carrying this specific CGD-causing variant. However, during this study, another patient at Aarhus University Hospital diagnosed with X-linked CGD was identified with a *CYBB* c.252 G > A variant leading to exon 3 skipping. As we were able to obtain PBMCs, as well as CD34^+ HSPCs from the patient's heterozygous healthy mother, we sought to use the optimized gene editing strategy to establish an analogous gene editing approach for *CYBB* c.252 G > A (Fig. 3a). We designed two potentially allele-specific sgRNAs targeting only the mutated allele (sg1 and sg2), as well as a sgRNA targeting both alleles (sg3). The activity of these sgRNAs was confirmed in K562 model cell lines, which revealed that sg1 was not allele-specific, as indels were observed in wild-type K562 cells (Supplementary Fig. 8a, b). We therefore focused on assessing sg2 versus sg3 in combination with a rAAV6 repair template designed to correct the *CYBB* c.252 G > A variant and install silent mutations to disrupt the

binding sites of all sgRNAs. This resulted in up to 20% HDR in CYBB c.252 G > A heterozygous PBMCs for sg3 that targets *CYBB* without allele specificity (Fig. 3b). However, limited HDR was observed using the allele-specific sg2. In CD34⁺ HSPCs from healthy donors, we observed higher HDR rates using sg3, reaching >50% HDR, whereas no gene modifications were observed using the allele-specific sg2, as would be expected in wild-type CD34⁺ HSPCs (Supplementary Fig. 8c, d). To increase HDR rates using the allele-specific sg2, we optimized the HDR template for this sgRNA, so that it would correct the c.252 G > A variant without installing the additional silent mutations, and used this repair template solely with sg2. With this optimized setup, we were able to obtain efficient HDR rates in heterozygous CD34⁺ HSPCs carrying the *CYBB* c.252 G > A variant (hereafter referred to as CYBB-HSPCs), reaching >25% for cells treated with sg2 RNP + AAV (repair template *without* silent mutations) and up to 60% for cells treated with sg3 RNP + AAV (repair template *with* silent mutations) (Fig. 3c). We could also successfully differentiate the gene-edited CYBB-HSPCs into granulocytes (Fig. 3d), which we could assay for ROS production. As suspected, when nucleofecting with sg3 RNP only, we were able to almost completely abolish ROS production, which was rescued upon treatment with both RNP and rAAV6 (Fig. 3e). Interestingly, the cells treated with the allele-specific sg2 RNPs and rAAV6 showed a slight increase in ROS+ cells, although this was not sufficient to reach statistical significance (Fig. 3e, Supplementary Fig. 8e, f).

### Targeted insertion of *CYBB* cDNA enables near-universal treatment for X-CGD

Whereas the gene editing strategies presented so far were designed to precisely correct specific gene variants, the cargo capacity of rAAV6 allows targeted insertion of larger DNA sequences, potentially allowing 'one-size-fits-all' gene editing strategies. To explore a more universal treatment strategy for correction of X-CGD, we designed an HDR template carrying a *CYBB* exon 3–13 cDNA that would potentially, in combination with targeted DNA cleavage using sg3-containing RNPs, correct all mutations occurring downstream of the sg3 cut site, including the *CYBB* c.252 G > A variant (Fig. 3f). To increase gene expression from the reconstituted gene, we included a WPRE sequence downstream of the inserted cDNA, as this has previously been shown to ensure proper expression from cDNA insertions[10]. To allow tracking of targeted integration events using flow cytometry, we also designed a repair template containing a GFP reporter gene, cDNA-P2A-eGFP (Fig. 3f). We estimate that this cDNA insertion strategy could potentially treat >86% of X-CGD patients[41] (Fig. 3g, Supplementary Fig. 9a–c). As expected, when CYBB-HSPCs were treated with sg3 RNPs and rAAV6 carrying the cDNA-eGFP repair template, we did not observe any GFP+ cells in treated HSPCs 72 h after treatment. However, upon myeloid differentiation, we observed up to 40% GFP+ cells on day 14 after initiation of differentiation, demonstrating that the integrated cDNA-eGFP cassette was indeed integrated and lineage-specifically expressed (Fig. 3h). We also determined the integration levels on the genomic level by ddPCR and found that the *CYBB* cDNA was integrated in 47% of the targeted alleles, with the integration levels maintained following differentiation into CD15+ myeloid cells (Fig. 3i). We also assayed differentiated granulocytes for ROS production and again found that sg3 RNP-only treatment led to the abolishment of ROS production, while the addition of the cDNA repair template rescued ROS production in >50% of the cells (Fig. 3j, Supplementary Fig. 8g), although we did observe a modest decrease in the median fluorescent intensity of rhodamine-123 in cells with cDNA integration (Supplementary Fig. 8h).

### High-fidelity Cas9 restores multilineage engraftment potential in immunodeficient NOG mice

To determine the engraftment potential of gene-edited *CYBB* c.252 G > A HSPCs, we transplanted edited HSPCs into sub-lethally irradiated immunodeficient NOG mice and compared the engraftment potential

of mock-treated cells to either sg2 RNP + AAV (c.252 G > A correction *without* silent mutations) or sg3 RNP + AAV (c.252 G > A correction *with* silent mutations). Sixteen weeks after transplantation, the mice were analyzed for human chimerism. As expected, mock-treated HSPCs showed the highest engraftment. Notably, HSPCs edited using the allele-specific sg2 also showed robust engraftment, reaching a mean of 34% human chimerism in the bone marrow (Supplementary Fig. 10a–d). The distribution of cell lineages in the mice was also comparable between mock and sg2 RNP + AAV (Supplementary Fig. 10e–g). In contrast, HSPCs treated with sg3 RNP + AAV showed <1% human chimerism, despite similar viabilities of edited cells prior to engraftment (Supplementary Fig. 10a). We additionally sorted out both CD45⁺ and CD34⁺ cells from the bone marrow of transplanted mice and found that contrary to our initial engraftment studies of *CYBA*-edited HSPCs, editing rates were maintained to a higher degree in the engrafted cells after 16 weeks, when using the optimized HDR protocol (Supplementary Fig. 10h vs Fig. 1i), although we still observed a loss of editing events in the human graft.

We speculated that the decrease in engraftment potential observed in HSPCs treated with sg3 could be due to adverse off-target editing. Therefore, we performed DISCOVER-seq to nominate potential off-target sites for sg3 using both standard (STD) Cas9 as well as a high-fidelity (HiFi) Cas9 (Fig. 4a). We also included the top 10 in silico predicted off-target (OT) sites for sg3 and validated all candidate OTs by high-throughput sequencing (Fig. 4b, Supplementary Fig. 11a). We detected indels at 6 of 17 candidate OT sites, with OT1 showing 77.5% indels, close to the level observed at the on-target site (Fig. 4b). Editing at these off-targets could be significantly limited by using a high-fidelity Cas9 (HiFi Cas9), which limited off-target editing to <5% for all six OTs while retaining nearly the same on-target efficacy (Fig. 4b). We therefore aimed at evaluating if using sg3 in combination with HiFi Cas9 as opposed to standard Cas9 would improve HSPC fitness. We edited CYBB-HSPCs using sg3 RNP + AAV to directly correct the *CYBB* c.252 G > A variant using standard Cas9 as well as HiFi Cas9 and measured editing rates, viabilities, and proliferation. We also included HSPCs edited using sg2 RNP + AAV with standard Cas9. We saw a significant increase in viability and proliferation of edited HSPCs when using HiFi Cas9 to an extent by which the cells performed almost identically to mock-treated HSPCs (Supplementary Fig. 11b). Additionally, we performed a methylcellulose-based CFU assay on the edited HSPCs and observed a 27% reduction in the CFU potential of HSPCs treated with sg2 RNPs with standard Cas9 compared to mock-treated HSPCs (Fig. 4c), in line with the decreased engraftment potential observed previously. Furthermore, HSPCs treated with sg3 and standard Cas9 showed an almost complete loss of CFU potential (80% reduction) compared to mock-treated HSPCs (Fig. 4c), whereas the use of HiFi Cas9 rescued the CFU potential significantly to a level that was comparable to sg2-treated HSPCs and led to a significantly higher number of total cells in the CFU assay (Supplementary Fig. 11c).

We then set out to determine if the rescue we observed in CFU potential would also be reflected in an increased ability to engraft in immunodeficient mice. We carried out engraftment of gene-edited CYBB-HSPCs as previously, using either standard Cas9 or HiFi Cas9. Additionally, the AAV dose was reduced twofold from 5000 to 2500 vg cell⁻¹ to further aid the engraftment potential of gene-edited HSPCs. We again analyzed the mice 16 weeks after transplantation and found that use of HiFi Cas9 rescued the multilineage engraftment potential, reaching 67.3% engraftment comparable to mock-treated HSPCs (Fig. 4d, e, Supplementary Fig. 12a, b). We also observed a higher engraftment potential of standard Cas9-treated HSPCs, reaching 21.8% engraftment, which we attributed to the twofold lower AAV dose. However, when sequencing the human graft in the bone marrow, we also observed a substantial loss of gene editing levels from 58.2% HDR in the input HSPCs to 17.8% HDR in the human graft within the bone marrow (Fig. 4f). This loss was notably less pronounced in HiFi

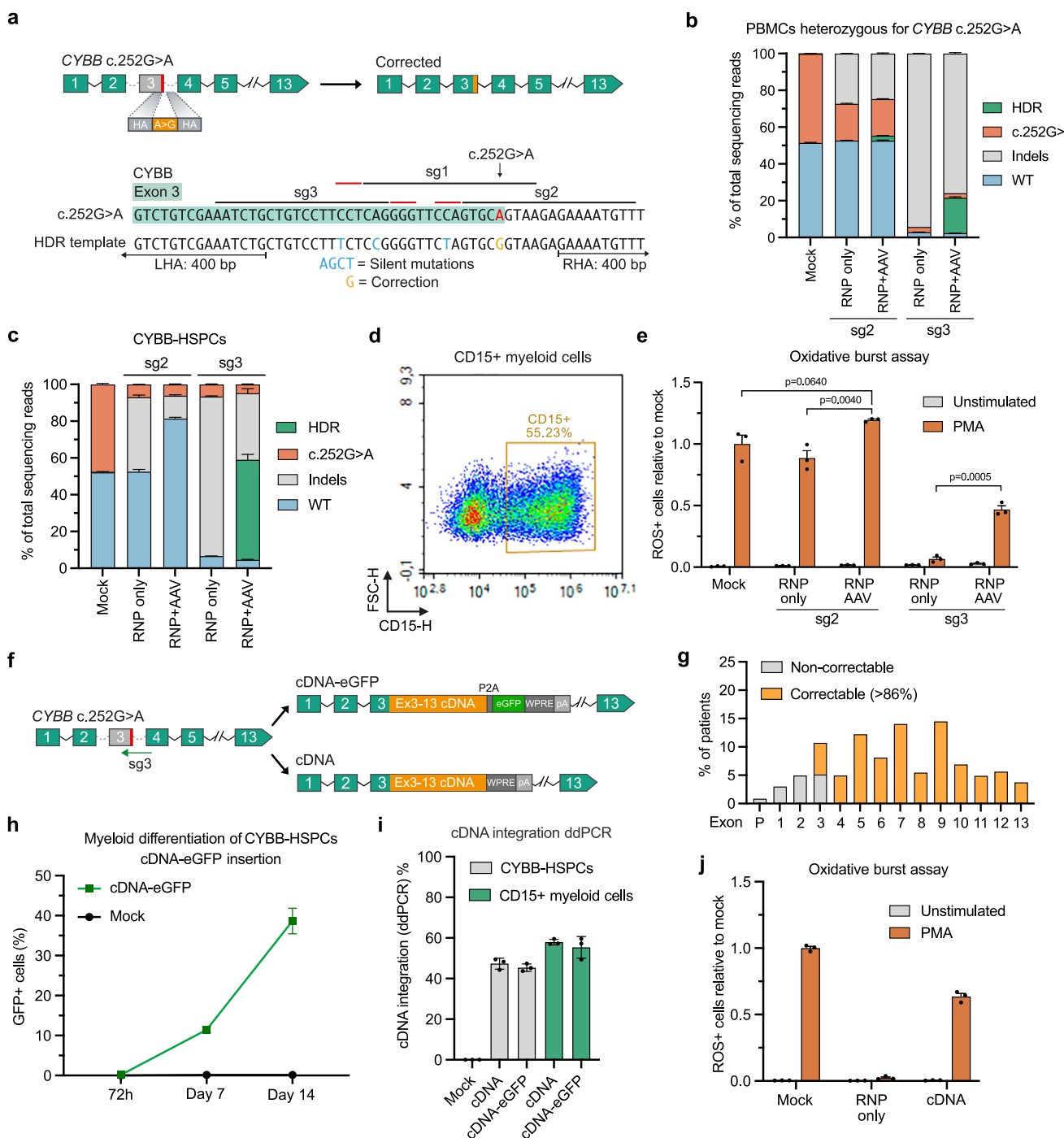

**Fig. 3 | Allele-specific editing and cDNA insertion for correction of *CYBB* variants. a** Schematic representation of the gene editing strategy and HDR template to correct the *CYBB* c.252 G > A variant. **b** RNP + AAV-mediated gene correction of *CYBB* c.252 G > A variant in heterozygous carrier peripheral blood mononuclear cells (PBMCs). **c** RNP + AAV-mediated gene correction of *CYBB* c.252 G > A variant in heterozygous carrier CD34+ HSPCs (CYBB-HSPCs) measured 4 days after nucleofection. The HDR template used for sg2-mediated correction installs a precise reversion of c.252 G > A to wild-type and thus, HDR events cannot be distinguished from wild-type alleles. **d** Representative plot of granulocyte differentiation as measured by the presence of CD15+ myeloid cells 14 days after initiation of differentiation of gene-edited CYBB-HSPCs. **e** Oxidative burst assay of granulocytes differentiated from gene-edited CYBB-HSPCs derived from a heterozygous carrier. **f** Schematic representation of the near-universal cDNA insertion strategies to

correct *CYBB* variants. **g** The percentage of X-CGD patients potentially covered by the cDNA insertion strategy. Calculations were performed based on Roos et al. [41]. The calculation of correctable patients does not take into account X-CGD cases caused by large deletions of part of chromosome X, such as variants found in patients with McLeod syndrome. **h** Targeted integration levels using the cDNA-eGFP repair template measured by GFP+ cells following granulocyte differentiation. **i** Targeted integration levels determined by ddPCR before and after myeloid differentiation. **j** Oxidative burst assay performed in granulocytes differentiated from heterozygous CYBB-HSPCs treated with sg3 RNPs and the cDNA repair template. Data represented as mean (*n* = 3 biological replicates) ± standard deviation. Statistical significance was determined by one-way ANOVA with Tukey's multiple comparisons test. Source data are provided as a Source Data file.

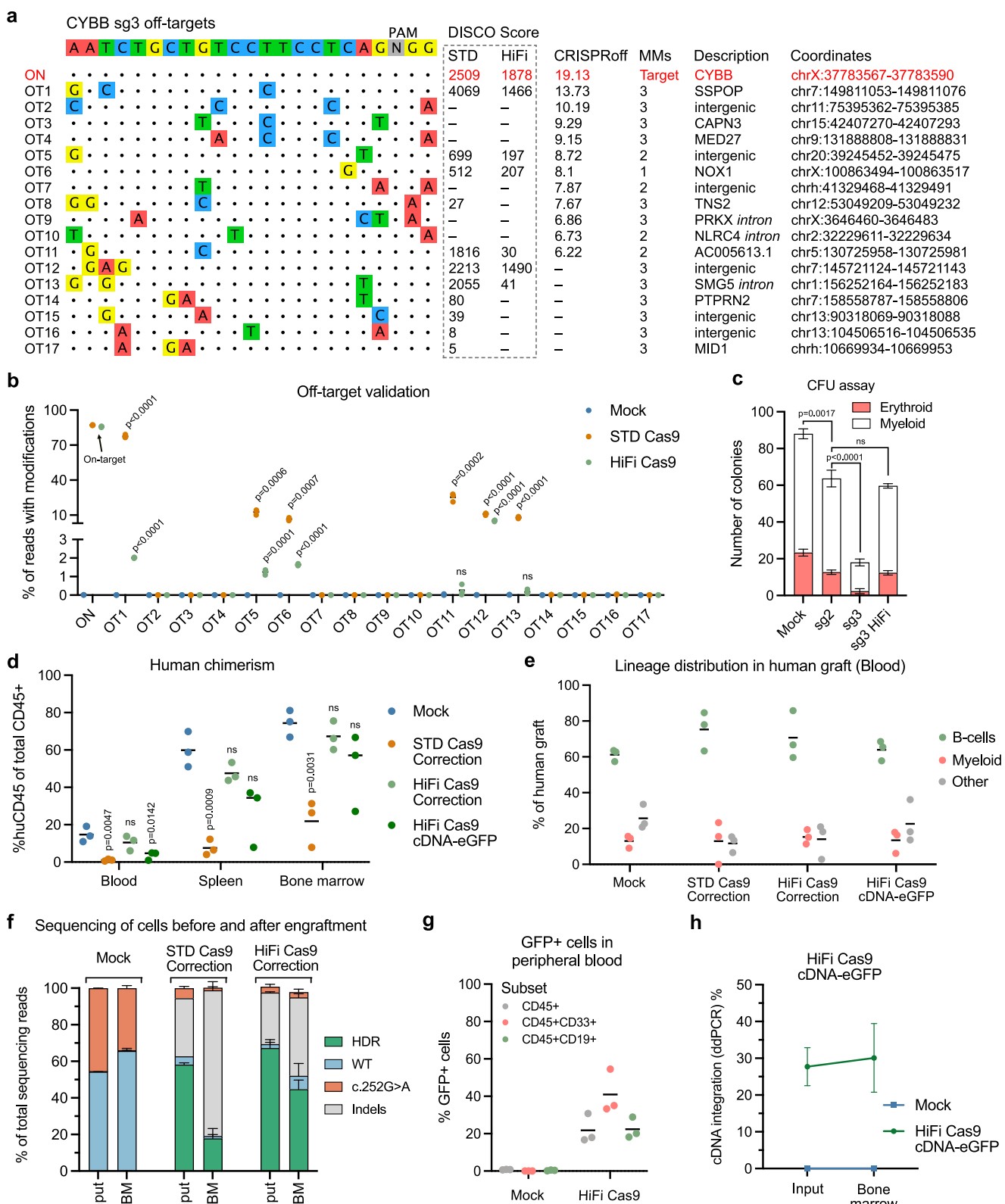

**Fig. 4 | HiFi Cas9 limits off-target editing and rescues the engraftment potential of gene-edited CYBB-HSPCs. a** Off-target sites of *CYBB* sg3 nominated by combining the top 10 in silico predicted off-target sites with DISCOVER-seq hits. DISCOVER-seq was carried out using both standard (STD) Cas9 and HiFi Cas9. **b** High-throughput amplicon sequencing of nominated off-target sites. **c** Semi-solid methylcellulose-based CFU assay of gene-edited CYBB-HSPCs. **d** Human chimerism in peripheral blood, spleen, and bone marrow of NOG mice 16 weeks after injection. **e** Distribution of human CD33⁺ myeloid cells and CD19⁺ B-cells in the peripheral

blood of NOG mice 16 weeks after injection. **f** Variant frequencies determined by sequencing of the human graft in the bone marrow (BM) of NOG mice, as well as of the injected CYBB-HSPCs (input). **g** The percentage of GFP⁺ cells within the human graft in the peripheral blood of NOG mice 16 weeks after injection. **h** ddPCR-based quantification of cDNA integration in the input CYBB-HSPCs and within the human graft in the bone marrow. Data represented as mean ($n = 3$ biological replicates) ± standard deviation. Statistical significance was determined by one-way ANOVA with Tukey's multiple comparisons test. Source data are provided as a Source Data file.

Cas9-treated HSPCs, which only showed a decrease from 67.6% HDR in the input HSPCs to 44.8% HDR in the human graft within the bone marrow of transplanted mice. For this round of engraftments, we also included CYBB-HSPCs that had been treated with HiFi Cas9 and the cDNA-eGFP repair template and observed that these engrafted comparably to HSPCs treated with HiFi Cas9 and the correction repair template, reaching 57.1% engraftment in the bone marrow (Fig. 4d, e). We could also detect GFP+ cells in all analyzed tissues, with 41% of peripheral blood CD45+CD33+ cells expressing GFP (Fig. 4g, Supplementary Fig. 12c, d). In line with the retained gene editing levels of cells treated with the correction repair template, we did not observe any loss of targeted integration of the cDNA-eGFP following engraftment (Fig. 4h).

Overall, these findings demonstrated the correction of an X-CGD-causing variant in heterozygous carrier HSPCs, although off-target editing with sg3 and standard Cas9 severely affected HSPC fitness as well as the engraftment potential of the edited HSPCs. Importantly, using HiFi Cas9 to restrict off-target editing rescued HSPC fitness, leading to rescue of multilineage engraftment potential with minimal loss of gene editing events 16 weeks after engraftment.

### Use of paired Cas9 nickases eliminates off-target editing and retains on-target efficacy

Despite the profound reduction in off-target editing observed when using HiFi Cas9, we still detected up to 4.5% off-target alterations at multiple sites, which challenged the safety of our gene editing strategy. Lastly, we therefore aimed to establish a gene editing strategy of CYBB using a Cas9 nickase (Cas9n) along with two paired sgRNAs, as this has previously been shown to limit off-target editing[42,43]. Based on previous work by others, we decided to use the D10A Cas9n variant with two PAM-out sgRNAs[44–47]. We designed two additional sgRNAs (sg4 and sg5) targeting the DNA strand opposite to sg3 upstream of the cut site (Fig. 5a) and found that the combination of sg3 and sg4 with D10A Cas9n led to HDR efficacies close to that of the already established sg3 HiFi Cas9 strategy (Fig. 5b, Supplementary Fig. 13a−e). As expected, the use of only sg3 with D10A Cas9n led to very low levels of HDR as was the case when sg3 and sg5 were used with D10A Cas9n (Fig. 5b). In line with previous reports of ITR insertion in gene-edited cells, we could detect ITR trapping events in the gene-edited CYBB-HSPCs, although the frequency of these were low (<0.1%) (Supplementary Fig. 14a). Notably, CYBB-HSPCs treated with sg3 + sg4 D10A RNPs showed an increase in proliferation compared to cells treated the sg3 HiFi RNPs (Fig. 5c), which were in line with a markedly reduced p53-response compared to HSPCs treated with sg3 standard Cas9 RNPs as indicated by a reduced fold change in p21 mRNA levels relative to mock (Fig. 5d). We subjected the gene-edited CYBB-HSPCs to a CFU assay and found again that the use of standard Cas9 led to a drastic decrease in the number colonies compared to mock-treatment. In contrast, using D10A Cas9n did not significantly lower the number of colonies from mock-treatment (Fig. 5e, Supplementary Fig. 14b). This indicated that the cytotoxicity of our D10A Cas9n gene editing strategy was reduced, although we noted a reduced size of the colonies using D10A Cas9n (Supplementary Fig. 14b).

The use of paired Cas9 nickases has previously been shown to decrease off-target editing[42,43,45,47], and indeed we also find that our paired D10A Cas9n strategy reduced all off-target editing to non-detectable levels (Fig. 5f). To assay for possible genomic recombinations, we additionally performed CAST-seq[43,48] of gene-edited CYBB-HSPCs and found that treatment of HSPCs with sg3 standard Cas9 RNPs led to 8 distinct off-target mediated translocations (OMTs) (Fig. 5g, Supplementary Fig. 15a) in line with the relatively high degree of off-target editing observed for this sgRNA (Fig. 4h). Using HiFi Cas9 reduced the number of OMTs to 5. Notably, CAST-seq did not identify any OMTs in cells treated with the sg3 + sg4 D10A Cas9n strategy. In addition, there was also a decrease in the percentage of CAST-seq

reads within ±10 kb of the sg3 cleavage site mapping more than 200 bp away from the target, indicating that the use of D10A Cas9n also decreased on-target large deletions as seen in the coverage plot (Fig. 5h, Supplementary Fig. 15b) and by quantification (Supplementary Fig. 15c, d).

Taken together, these results show that using paired nicking with D10A Cas9n allowed us to perform targeted gene editing of CYBB with retained on-target efficacy and minimal to no cytotoxicity. Our findings demonstrate the feasibility of therapeutic HDR-directed gene editing without detectable off-target editing and formation of chromosomal translocations.

## Discussion

Here, we develop and optimize CRISPR-Cas9-based gene editing strategies for correction of two CGD-causing variants identified in patients at Aarhus University Hospital in Denmark: a CYBA c.287+1 G > T variant (p22phox) causing autosomal CGD and a CYBB c.252 G > A variant (NOX2) causing X-CGD. We show potent gene correction of both CGD-causing variants and validate functional ROS production. We also show efficient insertion of a partial CYBB cDNA, which could constitute a near-universal treatment of an estimated 80% of X-CGD patients, and we develop a paired Cas9 nickase gene editing strategy of CYBB that does not lead to detectable off-target editing or chromosomal translocations.

For gene editing of CYBA in CD34+ HSPCs, we found that rAAV6 outperformed both ssODN and IDLV as a platform for HDR repair template delivery when comparing viabilities, proliferation, and editing rates. We found that both ssODN and IDLV had a slightly greater detrimental effect on HSPCs fitness compared to rAAV6 and, moreover, that these repair templates supported lower editing rates relative to rAAV6. This contrasts with recent studies that found less genotoxicity in HSPCs when using either IDLV or ssODN as compared to rAAV2/6 as a repair template[34,36]. However, it is possible that viability and proliferation are not sufficient fitness parameters to thoroughly characterize cytotoxic effects, although a DDR would be expected to influence both cell viability and proliferation. Nevertheless, additional assays to assess the DDR, CFU potential, and engraftment potential could allow a better assessment of HSPC fitness. In addition, a recent report found frequent concatemeric insertions of AAV vectors during Cas9- and AAV-mediated gene editing[49], which may potentially challenge the safety of HDR procedures based on rAAV6 repair templates. However, the authors also showed a mitigation strategy where the ITRs were cleaved from the repair template by including binding sites for the used sgRNA. Similarly, despite the lack of a functional integrase, IDLVs have also been shown to lead to vector integrations at sites of DSBs[34], with one report finding more off-target integrations from IDLV than AAV[50]. In line with these reports, we also find low levels of ITR trapping events in gene-edited CYBB-HSPCs. However, the potential impact of such trapping events will need to be investigated in more detail, along with the prevalence of larger concatemeric insertions, before moving on to clinical translation of gene editing strategies targeting CGD.

The negative impact of Cas9- and AAV-mediated editing in CD34+ HSPCs observed in this study in terms of viability, proliferation, and engraftment potential is in line with previous reports and has largely been attributed to the DDR initiated by both DSB formation and AAV transduction[32,34,37,51–54]. Additionally, HDR-mediated gene editing is thought to be less efficient in the primitive LT-HSCs, generally resulting in loss of edited cells following long-term engraftment. This is also evident in data from the now-discontinued Graphite Bio-led CEDAR trial, which shows an almost complete loss of gene correction after infusion into the patient[55]. Others have previously reported that the use of GSE56 and the adenoviral Ad5-E4orf6/7 proteins to inhibit p53-response and increase proliferation, respectively, alleviated clonogenic and repopulating

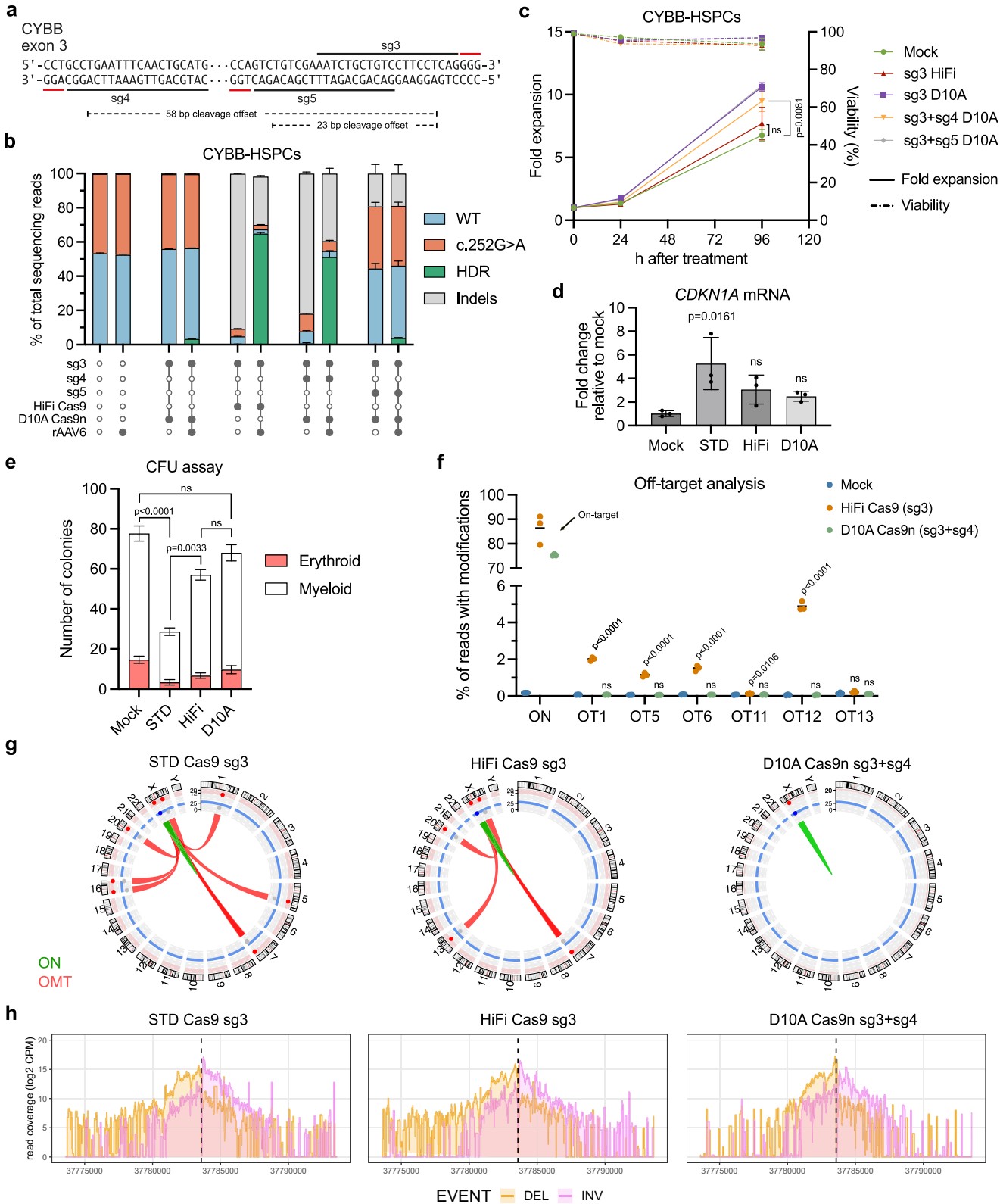

**Fig. 5 | Paired D10A Cas9n gene editing does not induce off-target editing and retains high on-target efficacy. a** Schematic of *CYBB* exon 3 showing binding sites for sg3, sg4 and sg5 as well as distance between D10A Cas9n-induced nicks. **b** D10A Cas9n-mediated gene correction of the *CYBB* c.252 G > A variant in CYBB-HSPCs measured 4 days after nucleofection. **c** Proliferation and viability data of RNP + AAV-treated cells from (**b**). **d** RT-qPCR quantification of p21 (*CDKN1A*) mRNA levels measured 48 h after nucleofection. **e** Semi-solid methylcellulose-based CFU assay of gene-edited CYBB-HSPCs. **f** Off-target analysis of CYBB-HSPCs treated with sg3 HiFi Cas9 RNPs or sg3 + sg4 D10A Cas9n RNPs. Only off-targets that showed detectable editing in Fig. 4b were assayed. **g** Circos plots showing off-target-mediated translocations (OMTs) identified by CAST-seq. **h** Coverage plots from CAST-seq showing all reads mapped to a ±10 kb region of the on-target site. Deletions (DEL) are shown in orange, and inversions (INV) are shown in purple. Data represented as mean (*n* = 3 biological replicates) ± standard deviation. Statistical significance was determined by one-way ANOVA with Tukey's multiple comparisons test. Source data are provided as a Source Data file.

disabilities of HSPCs[32,33]. Similarly, another study found that use of i53 to inhibit 53BP1 (a key protein in the non-homologous end-joining pathway) increased HDR rates in CD34+ HSPCs[10]. Here, we found that combining all three inhibitors/effectors led to the highest HDR rates in the LT-HSC-enriched CD34+CD45RA−CD90+ population, while still benefiting from increased viability and proliferation of gene-edited cells. We additionally showed that engrafted cells preserved editing rates 16 weeks after engraftment into immunodeficient mice, indicating that a combination of all three inhibitors/effectors allowed potent gene editing of LT-HSCs. In this study, we delivered both i53, GSE56, and Ad5-E4orf6/7 as mRNA, which, despite the inherent transient activity, was still expected to be expressed and present in the cells for multiple days. Given the role of these proteins on cell proliferation and DNA repair, a concern could be increased genomic instability, and shortening the duration of exposure to the modulators could potentially increase safety. As an alternative to mRNA, Baik et al. recently showed that delivery of i53 as a recombinant peptide limits the persistence of the protein in the cells, while still improving the long-term multilineage repopulating capacity of gene-edited HSPCs[56]. Yet another alternative to both mRNA- and peptide-based modulators of HDR is small-molecule inhibitors of DNA-PKcs. Of these, AZD7648 has previously been shown to supposedly increase HDR rates in HSPCs[57]. Importantly, however, a recent report found AZD7648 to cause excessively large deletions at the on-target site and to increase the formation of OMTs[58].

An important safety aspect of developing gene editing therapies is the potential for off-target editing. Indeed, in this study, we identified six prominent off-target sites for the *CYBB* sg3, of which OT1 showed indel formation at levels close to the on-target efficacy. *SSPOP* is a pseudogene encoding for subcommisural organ (SCO)-spondin and is thought to be expressed in the placenta during human development, but is weakly expressed in hematopoietic cells and has no reported function in the hematopoietic system[59]. Another validated OT, *NOX1*, encodes for NADPH oxidase 1 and is a homolog of NOX2, but contrary to NOX2, NOX1 is vaguely expressed in hematopoietic cells and primarily functions in intestinal epithelial cells[60–62]. However, despite the apparent lack of identified off-target sites that relate to hematopoietic function, we find a notable negative impact on the engraftment potential of HSPCs edited using *CYBB* sg3. Although we cannot rule out unknown off-target sites as the cause of this, we speculate that the impaired repopulating potential of the gene-corrected HSPCs reflects the excess formation of DSBs occurring in the treated cells. Indeed, when we subsequently used HiFi Cas9 to limit off-target DSB formation, we observed a reduction of off-target editing to <5%. Notably, although some high-fidelity Cas9 variants can be associated with a loss of on-target efficacy, we found that use of HiFi Cas9 did not result in a significant loss of editing efficacy, which is in line with the reported on-target efficacy of the R691A HiFi Cas9 variant[23]. Furthermore, by including HiFi Cas9, cell viability, proliferation, and CFU potential were significantly increased. This was also reflected in a rescue of engraftment potential to levels comparable to mock-treated cells, collectively indicating that limiting off-target editing as well as DSB formation reduced cellular toxicity significantly.

As an alternative to single sgRNA Cas9 nuclease-based gene editing, others have previously shown that using two paired sgRNAs with a D10A Cas9 nickase can support gene editing[42,44–47,58]. In this study, we developed a paired Cas9n strategy targeting *CYBB* that retained on-target gene editing efficacy but did not induce detectable off-target editing or lead to cytotoxicity. Furthermore, using CAST-seq, we identified fewer on-target aberrations and no detectable OMTs, highlighting the significantly improved safety of this gene editing strategy compared to previously developed *CYBB* gene editing strategies[10,13,63,64]. However, a D10A Cas9n-based gene editing strategy still depends on the formation of on-target DSBs and therefore benefits from the use of various modulators of HDR as well as p53 inhibition. Hence, the genome stability of treated CYBB-HSPCs will need to be investigated in the future to allow a clinical translation. In this regard, emerging DSB-free gene editing approaches, such as base and prime editing[65], are interesting alternatives for the treatment of CGD. Notably, however, recent reports have shown that these technologies do in fact generate low numbers of DSBs and induce a DDR in treated HSPCs[66]. Furthermore, these technologies are still patient-specific and need to be optimized for individual patients, hindering scalability to clinical use.

Overall, we report multiple gene correction strategies for both a *CYBA* c.287+1 G > T and a *CYBB* c.252 G > A variant. For both variants, we show potent gene correction, leading to functional ROS production in differentiated granulocytes. Importantly, the editing efficacies obtained in this study are well above the therapeutic threshold of approximately 20% ROS+ cells believed to be required for reconstitution of the neutrophil function in patients[40]. We also report a near-universal gene editing strategy for X-CGD by targeted integration of a truncated *CYBB* cDNA, covering 86% of X-CGD patients. It is important to note that while we found up to 47% targeted insertion of this truncated cDNA by ddPCR, the assay did not effectively discriminate between intact and partial integrations of the cDNA. Still, we found that the cDNA insertion led to functional ROS production in up to 60% of the cells relative to mock-treated cells. A limitation of the work presented here is that it was not possible for us to acquire patient-derived CD34+ HSPCs, and thus, we were only able to assay our gene editing strategy in heterozygous carrier CD34+ HSPCs from adult relatives. Given the chronic hyperinflammation often observed in CGD patients, the minimal cytotoxicity associated with the HiFi and D10A Cas9-based gene editing approaches reported here will need to be validated in patient HSPCs before clinical translation. Others have previously reported on gene editing strategies of *CYBB* as a treatment option for X-CGD[10,13,63,64]. However, the work presented here provides additional insights into the safety and off-target effects of HDR-directed gene editing strategies targeting *CYBB*, highlighting several key challenges that hinder the clinical translation of these gene editing-based therapies. Notably, we demonstrate the potential of a D10A Cas9n-based gene editing approach for *CYBB*, which shows retained high on-target efficacy, but which does not result in detectable off-target editing or OMTs. In combination with a near-universal cDNA insertion strategy, supporting targeted insertion of exon 3 through 13, such a knock-in gene editing strategy could potentially facilitate treatment of more than 86% of X-CGD patients. In conclusion, our results expand upon gene editing strategies targeting *CYBB* by demonstrating a promising curative treatment option for patients with X-CGD using paired D10A Cas9n-mediated genome editing with no detectable off-target editing or chromosomal rearrangements.

## Methods

### Ethics statement

All studies performed in this work comply with all relevant ethical regulations. Collection and experimental work on CD34+ HSPCs were performed under a study protocol approved by the Central Denmark Region Committees on Health Research Ethics with approval number 1-10-72-144-19. Written consent was obtained from all HSPC donors. Animal experiments were performed under the approval of The Danish Animal Inspectorate (License no. 2018-15-0201-01506 and 2023-15-0201-01458).

### CD34+ HSPC purification and culture

Mobilized peripheral blood (mPB) CD34+ HSPCs were collected by standard procedures after informed consent from healthy donors (N = 4) and a heterozygous carrier (N = 1) at the Department of Clinical Immunology, Aarhus University Hospital, under a study protocol

approved by The Central Denmark Region Committees on Health Research Ethics with approval number 1-10-72-144-19. Study data were collected and managed using REDCap electronic data capture tools hosted at Aarhus University, Aarhus, Denmark[67,68]. CD34+ HSPCs were purified from apheresis products using a CliniMACS Prodigy (Miltenyi Biotec), using the "CD34 enrichment" standard procedure, according to the instructions by the manufacturer. Purified CD34+ HSPCs were subsequently cryopreserved and stored until use. Self-reported sex and/or gender, number, and age of all HSPC donors can be found in Supplementary Table 7. Age, sex, and/or gender were not considered in the study design, and all experimental work was conducted and presented without regard to these variables. For all experiments, cryopreserved CD34+ HSPCs were thawed and seeded at a density of $1 \times 10^5$ cells mL$^{-1}$ in StemSpan™ Serum-Free Expansion Medium II (SFEM II, STEMCELL Technologies) supplemented with 100 ng mL$^{-1}$ Flt3-Ligand, 100 ng mL$^{-1}$ human SCF, 100 ng mL$^{-1}$ human TPO, 30 ng mL$^{-1}$ human IL-6, 30 ng mL$^{-1}$ human IL-3, 35 nM UM171, 1 µM StemRegenin 1 (SR1), and 20 U mL$^{-1}$ penicillin/streptomycin (P/S). For all experiments, CD34+ HSPCs were pre-expanded for 72 h prior to nucleofection.

## Cell lines and culture

HEK293T cells (ATCC, #CRL-3216) were maintained in DMEM supplemented with 5% fetal bovine serum (FBS) and 1% penicillin/streptomycin (P/S). K562 cells (ATCC, #CCL-243) were maintained in RPMI-1640 medium (Sigma–Aldrich) supplemented with 10% FBS and 1% P/S. To generate K562 cell lines harboring the *CYBB* c.252 G > A variant, K562 cells were nucleofected with recombinant Cas9 and an sgRNA targeting the wild-type *CYBB* allele (gRNA_c.252 G > A (Synthego) along with an ssODN (Integrated DNA Technologies, IDT) designed to install the c.252 G > A variant (CYBB_c252GA>A_s-sODN). Clones were subsequently generated by single-cell seeding and expansion. Genotypes were verified by Sanger sequencing (Eurofins Genomics). All cells were incubated, maintained, and cultured at 37 °C with 5% $CO_2$.

## HDR repair template design and construction

HDR repair templates were constructed as described previously[8]. Unless otherwise specified, repair templates contained homology arms of 400 bp flanking the desired correction sequence. For all repair templates, except the repair template used with the *CYBB* sg2, silent mutations were included to hinder re-cutting by Cas9 after HDR. To generate cDNA HDR repair templates, codon-diverged cDNA fragments were ordered from Twist Bioscience. Single-stranded oligonucleotides (ssODNs) were 100–200 nt in length and ordered as Alt-R HDR Donor Oligos (IDT). Repair templates for rAAV6 and IDLV delivery were assembled using NEBuilder HiFi DNA Assembly (New England Biolabs, NEB) into a pAAV-backbone containing AAV2-specific ITRs and pCCL vector backbones, respectively, following the manufacturer's instructions. Sequences of sgRNAs and HDR templates are available in Supplementary Tables 1 and 2, respectively.

## Viral vector production, purification, and titration

Recombinant AAV2/6 (rAAV6) was produced by iodixanol gradient purification, using previously published protocols[8,69]. Briefly, $11 \times 10^6$ HEK293T cells were transfected with 6 µg AAV repair template plasmid and 22 µg of the helper plasmid pDGM6 using PEI (4:1 PEI to DNA ratio) and cultured in DMEM with 1 mM Sodium Butyrate. AAV preps for correction of *CYBB* variants were produced in Gibco Viral Production Cells 2.0 (Thermo Fisher Scientific, #A51218) and transfected with AAV-MAX (Thermo Fisher Scientific), following the manufacturer's protocol. For both cases, AAV particles were harvested 3–4 days after transfection by three freeze-thaw cycles followed by incubation with 200 U mL$^{-1}$ Turbonuclease (Sigma–Aldrich). AAV vectors were subsequently purified by ultracentrifugation (490,000 x *g*, 10 °C, 90 min)

through an iodixanol gradient (OptiPrep, STEMCELL Technologies), consisting of a 15%, 25%, 40%, and 58% layer. AAV particles were extracted from the interface between the 40% and 58% layer and diluted in PBS containing 5% sorbitol and 0.001% pluronic acid, before finally concentrating particles using a 100 kDa MWCO Amicon Ultra Centrifugal Filter (Merck). The viral titer of AAV preps was quantified by a multiplex ddPCR assay, using primer/probes targeting the ITRs and bGH poly-A sequence when applicable[70,71]. Briefly, a 5 µL aliquot of the AAV preps was digested with 10 U DNase for 30 min followed by lysis using QuickExtract (Lucigen). Preps were then diluted 10.000-fold to 500.000-fold, from which 5 µL was used as input for ddPCR with SuperMix for Probes (No dUTP) (Bio-Rad). Reactions were analyzed on a QX200 Droplet Reader (Bio-Rad). AAV titers were ultimately calculated as $\frac{vg}{µL} = X \frac{copies}{µL} * 5µL * D$, where vg µL$^{-1}$ is the AAV titer, X is the copies µL$^{-1}$ readout from the ddPCR, and D is the total dilution factor.

IDLVs were produced using calcium-phosphate transfection of Lenti-X 293 T cells as described elsewhere[72]. In short, $4 \times 10^6$ Lenti-X 293 T cells were co-transfected in p10 dishes with 3 µg pRSV-Rev, 3.75 µg pMD.2 G, 13 µg pMDLg/pRRE-D64V, and 13 µg pCCL plasmid containing the HDR template. The supernatant was harvested 48 and 72 h after transfection, filtered through a 0.45 µm filter, and concentrated by ultracentrifugation at 100,000 x *g* (4 °C) through a 20% sucrose cushion. The IDLV was then resuspended in PBS and centrifuged at 2000 x *g* for 2 min to remove debris. IDLV titers were determined using a HIV-1 p24 ELISA (XpressBio) and back-calculating estimated IU mL$^{-1}$ following the manufacturer's instructions.

## In vitro transcription of mRNA

Templates for in vitro transcription were generated by inserting i53, GSE56, and Ad5-E4orf6/7 coding sequences into plasmids containing a T7 promoter, a 3' UTR derived from the murine *Hba* gene, and a 50 nt poly-A tail. Plasmids were then linearized using BbsI (NEB) to allow run-off transcription and purified by precipitation using 5 M $NH_4$ acetate and ethanol. In vitro-transcribed mRNA was produced with MegaScript T7 Transcription Kit Plus with full substitution of UTP with $N_1$-methylpseudouridine-5'-triphosphate (Tebubio) and co-transcriptional capping with CleanCap Reagent AG (TriLink Bio-Technologies) following the manufacturer's instructions. Transcribed mRNAs were precipitated in 2.5 M lithium chloride, washed once in 70% ethanol, dissolved in TE buffer, and quantified using a UV-Vis spectrophotometer.

## Gene editing of cell lines and human CD34+ HSPCs

For gene editing experiments, cells were nucleofected with 6 µg Cas9 (Alt-R™ S.p. Cas9 Nuclease V3 or Alt-R™ S.p. HiFi Cas9 Nuclease V3, IDT) and 3.2 µg synthetic sgRNA (Synthego) using a Lonza 4D-Nucleofector X (Lonza). Cells were washed in Gibco Opti-MEM Reduced Serum Medium (K562 cells, Thermo Fischer Scientific) or PBS (CD34+ HSPCs) and resuspended in 18-20 µL 1 M buffer (50 mM D-Mannitol, 5 mM KCL, 120 mM $Na_2HPO_4$, 15 mM $MgCl_2$) prior to nucleofection. When applicable, 100 pmol ssODNs (IDT) were included in the nucleofection. K562 cells were nucleofected using the CM-138 program and the Primary Cell P3 setting. CD34+ HSPCs were nucleofected using the DZ-100 program and the Primary Cell P3 settings. When i53, GSE56, and/or Ad5-E4orf6/7 mRNAs were used, 3 µg of each was included in the nucleofection. Mock nucleofections were carried out the same way, but without adding sgRNA, Cas9, or mRNA. Following nucleofection, cells were seeded at $5 \times 10^5$ cells mL$^{-1}$ and transduced with viral vectors carrying HDR repair templates within 15 min. For transductions using rAAV6, cells were initially seeded at a density of $5 \times 10^5$ cells mL$^{-1}$ and transduced with rAAV6 for 16 h before diluting cells to $1 \times 10^5$ cells mL$^{-1}$. A AAV dose of 5000 vg cell$^{-1}$ was used, unless otherwise specified. For IDLV transductions, two different protocols were used (Supplementary Fig. 6a). One protocol used plates coated with RetroNectin (Takara Bio) and LentiBoost (Sirion Biotech) as

transduction boosters, following the manufacturer's protocols. For this, cells were seeded at $5 \times 10^5$ cells mL$^{-1}$ in RetroNectin-coated plates and transduced using spinoculation ($800 \times g$, 32 °C, 40 min) with IDLV at an MOI of 300 for 16 h in the presence of LentiBoost prior to nucleofection. Cells were subsequently seeded at $1 \times 10^5$ cells mL$^{-1}$ after nucleofection without additional transductions. The other protocol used Cyclosporin H (CsH) as a transduction enhancer, following a previously published protocol[34]. Briefly, HSPCs were transduced 16 h prior to nucleofection at an MOI of 150 in the presence of 8 μM CsH and subsequently subjected to another round of transduction after nucleofection (MOI 150, 8 μM CsH), again at a density of $5 \times 10^5$ cells mL$^{-1}$. 16 h after the last transduction, the cells were resuspended in fresh medium to a density of $1 \times 10^5$ cells mL$^{-1}$. All cells were cultured for a minimum of 72 h after nucleofection before analysis of gene editing.

### Flow cytometry and FACS

Fluorescence was measured on a NovoCyte 2100 YB Analyzer (Agilent) equipped with two lasers (488 nm and 561 nm) or a NovoCyte Quanteon 4025 flow cytometer (Agilent) equipped with four lasers (405 nm, 488 nm, 561 nm, and 637 nm). The acquired data were analyzed using NovoExpress v. 1.5.6 or FlowJo v. 10.8.2. Flow cytometry-assisted cell sorting (FACS) was performed on a Bigfoot cell sorter (Thermo Fisher) equipped with six lasers (349 nm, 405 nm, 488 nm, 561 nm, and 640 nm). Sorting data were analyzed using SQS v.1.9.4. Stainings were performed in suitable buffers for 30 min at 4 °C using titrated antibodies. A list of antibodies and reagents used for flow cytometry can be found in Supplementary Table 3.

### Granulocyte differentiation and oxidative burst assay

To differentiate edited CD34$^+$ HSPCs to granulocytes, cells were seeded in SFEM II supplemented with StemSpan™ Myeloid Expansion Supplement (STEMCELL Technologies) 48 h after nucleofection and expanded following the manufacturer's instructions. The differentiation was monitored and validated by loss of CD34 expression and gain of CD15 expression. 14–16 days after initiating differentiation, the cells were assayed for ROS production by performing an oxidative burst assay. Briefly, $0.5–1 \times 10^5$ cells were resuspended in 100 μL SFEM II medium containing 500 ng Dihydrorhodamine 123 (DHR-123, Sigma–Aldrich) and 2.5 μM Phorbol 12-myristate 13-acetate (PMA, STEMCELL Technologies), and incubated at 37 °C for 25 min. Cells were additionally stained with a PE-conjugated anti-human CD15 antibody. ROS production was subsequently assayed in CD15$^+$ cells by fluorescence on a NovoCyte 2100 YB flow cytometer.

### Colony-forming unit assays

We performed colony-forming unit assays using two different methods. For Fig. 1, we performed a flow cytometry-based CFU assay, using StemMACS HSC-CFU Assay Kit (Miltenyi Biotec) following the manufacturer's instructions. Briefly, we edited CD34$^+$ HSPCs using our standard protocol, and 48 h after nucleofection, we seeded the cells in 96-well plates at a density of 2.5 cells/well. Plates were then placed in a humidity chamber for 14 days, at which point the cells were stained and analyzed on a NovoCyte Quanteon flow cytometer. Colonies containing >500 cells were scored based on expression of CD15, CD14, and CD235a, following general guidelines from Miltenyi. For Figs. 4 and 5, we performed semi-solid methylcellulose-based CFU assays using MethoCult™ H4435 Enriched (STEMCELL Technologies), following the manufacturer's instructions. In short, 48 h after nucleofection, 400 HSPCs were seeded in 1.1 mL MethoCult medium in 35 mm dishes using a 3 mL syringe with a 16 G needle. The cells were then placed in a humidity chamber for 14 days, before counting and scoring colonies. Colonies were scored as either erythroid or myeloid following general guidelines.

### RT-qPCR

Activation of p53 was determined by assaying mRNA levels of CDKN1A (p21), a downstream target of p53[73], using TaqMan Gene Expression assays and GAPDH to normalize gene expression levels. Total RNA was extracted from treated CYBB-HSPCs 48 h after nucleofection using phenol:chloroform extraction followed by purification using Roche High Pure miRNA Isolation Kit (Roche). The purified RNA was treated with DNase before being used in a two-step Maxima RT-qPCR (Thermo Fisher), following the manufacturer's instructions. Briefly, first-strand cDNA synthesis was carried out using Maxima cDNA H Synthesis Master Mix with 250 ng template RNA as input. The cDNA reactions were then diluted 6×, from which 4.5 μL were used as input in a 10 μL Maxima Probe qPCR Master Mix (no ROX) reaction, which was run on a LightCycler 480 in technical duplicates (Roche), following the manufacturer's guidelines.

### High-throughput amplicon sequencing and analysis

To assess gene editing outcomes in CD34$^+$ HSPCs, the targeted loci were amplified from genomic DNA using a three-step PCR protocol. Briefly, the first PCR (PCR1) amplified the genomic region of interest with at least one primer binding outside the homology arms of the HDR templates. The second PCR (PCR2) attached Illumina TruSeq sequencing adapters. The third PCR (PCR3) added indexes along with the P5 and P7 sequence adapters. PCR1 was run on either 5 μl QuickExtract lysate or 660 ng purified gDNA in a 50 μL Phusion Plus reaction (Thermo Fisher Scientific) for 28 cycles and purified by excision from a 1% agarose gel (Omega Bio-Tek) to ensure removal of residual DNA repair template. PCR2 was run on 10 μL purified PCR1 product in a 25 μL Phusion Plus reaction for 8 cycles. PCR3 was run on 4 ng purified PCR2 product for 8 cycles. All amplicons were purified with the HighPrep™ PCR Clean-up System (MagBio) and quantified using a Qubit fluorometer with the 1× dsDNA HS Assay Kit (Thermo Fisher Scientific). Sequence-ready amplicons were then pooled and sequenced with 150-bp paired-end reads on an Illumina iSeq 100 or MiniSeq (Illumina). Editing rates were determined with CRISPResso2 in HDR mode using standard parameters[74].

### Quantification of cDNA integration levels

Targeted integration of the *CYBB* cDNA was determined by digital droplet PCR (ddPCR). Genomic DNA from QuickExtract lysates was purified with the HighPrep™ PCR Clean-up System (MagBio) and subsequently digested with HindIII (Thermo Fisher Scientific) for 1 h at 37 °C. 5 μL HindIII-digested DNA was subsequently used in a 2-dimensional ddPCR reaction using the ddPCR Supermix for Probes (no UTP) and analyzed on the QX200 droplet digital PCR system (Bio-Rad). The assay used a custom primer/probe set amplifying a 466-bp amplicon, where the forward primer binds outside of the left HA, while the reverse primer binds in the codon-diverged *CYBB* cDNA to ensure amplification of only alleles with the cDNA integrated (Supplementary Table 4). Normalization was done using a primer/probe targeting *ALB* (albumin). Targeted integration levels were calculated as:

$$\text{integration\%} = \frac{\text{cDNA}^{CYBB} \frac{\text{copies}}{\mu L}}{\text{ALB} \frac{\text{copies}}{\mu L}} * 100 \tag{1}$$

### Identification of off-target sites with DISCOVER-seq

DISCOVER-seq was carried out as previously described[75,76]. Briefly, $2 \times 10^7$ K562 cells were nucleofected with either mock, sg3/STD-Cas9 RNPs, or sg3/HiFi-Cas9. 12 h after nucleofection, the cells were crosslinked in 1% formaldehyde for 15 min, washed in PBS, and pelleted. The pellets were subsequently resuspended in 10 mL LB1 and incubated for 10 min on ice before being spun down at $1000 \times g$ for 3 min at 4 °C. The pellets were then resuspended in 10 mL LB2 and

incubated on ice for 5 min before being spun down and resuspended in 1 mL LB3. The nuclear extracts were then sonicated using a Q800 sonicator (Qsonica) (Amplitude: 70%, Pulse Rate: 15 s on/45 s off, Total Sonication On Time: 40 min, Temperature: 4 °C). Hereafter, the samples were spun down at 17.000 × *g* for 15 min to pellet debris. The supernatant was mixed with 1.85 mL LB3 and 150 μL 20% Triton X-100 and mixed with anti-MRE11-bound Dynabeads™ Protein A (Thermo Fisher Scientific). The beads were then incubated, rotating at 4 °C for 16 h, after which the beads were washed 5 times in RIPA buffer and once in TBS before resuspending in 200 μL elution buffer. The crosslink was subsequently reversed by 6 h of incubation at 65 °C. Hereafter, 100 μL TE buffer was added to the samples before being treated with RNAse A (Thermo Fisher Scientific) and, hereafter Proteinase K (Thermo Fisher Scientific). The DNA was then purified from the samples using the MinElute PCR Purification Kit (Qiagen). The samples were library-prepped using the NEBNext Ultra II DNA Library Prep Kit for Illumina (New England Biolabs) and sequenced with 50 million 150 bp PE at the AUH NGS core on a NovaSeq™ 6000. Reads were mapped using BWA and analyzed using BLENDER2 with default filtering options. Complete DISCOVER-seq output can be found in Supplementary Data 1.

### Identification of translocations with CAST-seq

Genomic DNA was extracted using the DNeasy Blood & Tissue Kit (Qiagen) according to the manufacturer's protocol. Primer sequences used for CAST-seq are listed in Supplementary Table 4. CAST-seq analyses were performed as originally described for single-nuclease setting[48], and the D-CAST-seq bioinformatic pipeline for the use of the dual nickase[43]. In brief, 500 ng of genomic DNA was used as input material for each technical replicate. Libraries were prepared using the NEBNext® Ultra™ II FS DNA Library Prep Kit for Illumina (NEB). Enzymatic fragmentation of the genomic DNA was aimed at an average length of 500–700 bp. CAST-seq libraries were sequenced on a Next-Seq6000 using 2 × 150 bp paired-end sequencing (GENEWIZ, Azenta Life Sciences). For each sample, 3 technical replicates were run and analyzed. Only sites that were present in two technical replicates and significant in at least two replicates are reported (Supplemental Data 2). The presence of large aberrations (deletions and inversions >200 bp) was calculated as a percentage of the total on-target reads. Calculations were based on the read coverage in a ±10 kB window around the on-target site, with only the first base located closest to the on-target site of each read included in the computation. The percentage of large aberrations is equal to the sum of all reads with a start position >200 bp from the cleavage site, divided by the sum of all reads at the on-target site. Note that this value is relative to all CAST-seq reads at the on-target site and is not an absolute quantification of on-target aberrations. The mean deletion length was calculated based on the same read coverage:

$$\text{mean deletion length(bp)} = \frac{\text{sum(count of deletion reads*distance from cleavage site)}}{\text{sum of all deletion reads}} \quad (2)$$

Deletion reads are all negative reads because of the primer orientation for CAST-seq library preparation.

### Off-target nomination and validation

Potential off-targets for *CYBB* sg3 were nominated by DISCOVER-seq as described above or by in silico prediction using CRISPRoff[77]. For CRISPRoff, the top 10 off-targets with ≤3 mismatches were chosen for validation. Validation was carried out by targeted high-throughput amplicon sequencing. Sequencing was carried out as described above; however, a two-step PCR protocol was used, where PCR1 amplified the off-target of interest and attached TruSeq adapters. PCR2 then attached indexes as well as P5 and P7 adapters. PCR1 was run for 28 cycles, and PCR2 was run for 8 cycles. Data were analyzed using CRISPResso2 with the -ignore_substitutions parameter included to filter away sequencing errors in the quantification. CRISPResso2 results without the ignore_substitutions parameter can be found in Supplementary Fig. 11a.

### Transplantation of HSPCs into NOG mice

All animal experiments were performed under the approval of The Danish Animal Inspectorate (License no. 2018-15-0201-01506 and 2023-15-0201-01458). Mice were kept on a 12 h/12 h light/dark cycle at the Animal Facilities at the Department of Biomedicine, Aarhus University, Denmark. Mice had ad libitum access to Altromin maintenance feed (Altromin) and water. Mice were maintained and weighed weekly throughout the experiments. To assess the engraftment potential of gene-edited CD34⁺ HSPCs and the level of gene editing in LT-HSCs, we transplanted gene-edited HSPCs into 6–8-week-old immunodeficient NOG female mice (NOD.Cg-*Prkdc^{scid} Il2rg^{tm1Sug}*/JicTac, Taconic Biosciences). For each mouse, we treated $2 × 10^6$ CD34⁺ HSPCs using methods described earlier. 48 h after nucleofection, all outgrowth was collected and resuspended in 210 μL PBS before tail-vein injection into sub-lethally irradiated NOG mice (75–150 cGy). Mice were irradiated 24 h prior to injection. A small aliquot of cells was kept in culture to assess gene editing levels in the transplanted cells. The total number of transplanted cells is reported in Supplementary Table 5 and 6. 16 weeks after injections, all mice were euthanized, and blood, spleen, and bone marrow were collected for analysis of engraftment by flow cytometry. Engraftment levels were determined as the percentage of human CD45⁺HLA-ABC⁺ cells within the CD45⁺/CD45.1⁺ population. We additionally sorted out human cells from the bone marrow to assess gene editing levels in the engrafted cells. The gating strategy is available in Supplementary Fig. 3.

### Statistics and reproducibility

Unless otherwise specified, all data are represented as the mean of 3 independent biological experiments. Sample sizes were chosen based on cell and animal availability, as well as previous studies and generally accepted guidelines in the field. No statistical method was used to predetermine sample size. For engraftment studies, NOG mice were randomly chosen to receive cells from a given treatment. The Investigators were not blinded to allocation during experiments and outcome assessment. No data were excluded from the statistical analyses. Statistical analysis was performed using GraphPad Prism version 10.0.3. The statistical tests used are listed in the figure legends.

### Reporting summary

Further information on research design is available in the Nature Portfolio Reporting Summary linked to this article.

## Data availability

All data supporting the findings of this study are available within the paper and its supplementary information files. Raw sequencing data of CD34⁺ HSPCs are deposited at the European Genome-Phenome Archive (EGA), which is hosted by the European Bioinformatics Institute and the Center for Genomic Regulation. The data are available under the accession number EGAS50000001155 under controlled access at EGA due to privacy laws and legal restrictions associated with sharing sensitive data under the General Data Protection Regulation (GDPR). DISCOVER-seq data generated in this study have been deposited in NCBI's Gene Expression Omnibus and are accessible through the GEO Series accession number GSE287370. Source data are provided with this paper.

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

## Acknowledgements

We thank Charlotte Thornild Møller for excellent technical support. Flow cytometry and FACS were performed at the FACS Core Facility, Aarhus University, Denmark. J.H.W. and T.W.S. were funded by PhD fellowships provided by the Graduate School of Health at Aarhus University. This study was conducted as part of the PASCAL-MID research center (Personalized Approach to Genome Editing of Stem Cells for Autotransplantation of Monogenic Immunodeficiencies) funded by the Innovation Fund Denmark (Grant: 8056-00010 A). Additional funding was provided by the Novo Nordisk Foundation (NNF22OC0080684) and the German Research Foundation (DFG) to T.C. (CA 311/4-1 and FANEDIT/EJPRD20-209). Funding of the Bigfoot Cell Sorter was provided by the Carlsberg Foundation (personal grant to J.G.M.; ID CF21-0363). This study was also supported by Snedkermester Sophus Jacobsen og hustru Astrid Jacobsens Fond and Grosserer L.F. Foghts Fond.

## Author contributions

J.G.M. and R.O.B. conceived and supervised the project. J.H.W., T.W.S., S.R.D., and D.H. designed, performed, and analyzed experiments. D.H. and A.S.R. purified CD34+ HSPCs. B.K.M. supervised immunological and molecular diagnostics of patients and recruited voluntary donors. T.H.M. and M.H. saw patients, and T.H.M., M.H., and S.E.J. collected samples from patients. J.H.W., T.W.S., M.K.T., and D.H. performed animal experiments. T.C., S.A., and C.A. facilitated and performed CAST-seq. J.H.W. and J.G.M. wrote the manuscript. All authors read and approved the manuscript.

## Competing interests

The authors declare the following competing interests: J.G.M. is a member of the Scientific Advisory Board of nChroma Bio. R.O.B. is a co-founder of and consultant to UNIKUM Tx and is co-inventor on patent number WO2016164356A1, titled "Chemically modified guide RNAs for CRISPR/Cas-mediated gene regulation" co-filed by The Board Of Trustees Of The Leland Stanford Junior University and Agilent Technologies, Inc. with the following inventors: Matthew H. Porteus, Ayal Hendel, Joe

Clark, Rasmus O. Bak, Daniel E. Ryan, Douglas J. Dellinger, Robert Kaiser, and Joel Myerson. The patent is granted and relates to the use of chemically modified guide RNAs used in the manuscript. R.O.B. reports research funding from Novo Nordisk. None of the companies were involved in the present study. The remaining authors declare no competing interests.
