## [Transparent Peer Review file · Nature Communications]

Targeted gene editing and near-universal cDNA insertion of CYBA and CYBB as a treatment for chronic granulomatous disease

Corresponding Author: Professor Jacob Mikkelsen

Version 0:

Reviewer comments:

Reviewer #1

(Remarks to the Author)

In this manuscript, Wolff and colleagues describe the use of different strategies to correct individual mutations causing chronic granulomatous disease (CGD) and replace the missing genes in hematopoietic stem cells. They evaluate the use of single stranded DNA, AAV and IDLV-based templates for gene correction/replacement in HSCs. They evaluated the efficacy of gene correction using the different approaches and their impact on the colony forming and engraftment potential of the edited HSCs. They also compared the impact of using high fidelity Cas9 against wild-type Cas9 on the colony forming potential of the HSCs.

The results in figure 1 show that although the use of Cas9 with AAV templates enabled efficient gene insertion/correction in the CYBA locus, the edited HSCs showed fewer colonies as well as reduced engraftment compared to unedited controls. The results in Figure 2 show that the addition of mRNA encoding factors previously reported to improve HR, only improved HR modestly. Although they attempted gene insertion using alternative templates (ssODN and IDLV), AAV6 templates resulted in the most efficient gene correction. In addition, the corrected HSCs produced granulocytes with restored ROS+. Figure 3 shows the correction of mutations in CYBB and the replacement of the partial cDNA of CYBB. Figure 4 presents data showing that edited HSCs show poor engraftment potential. Notably, colony forming capacity of the HSCs edited using Cas9 and AAV seems to be restored when high fidelity Cas9 is used. The methodology is sound and the manuscript is written well.

The correction of CYBA mutations and the restoration of ROS function appears to be a new finding. However, other studies have previously reported the replacement of the CYBB cDNA from exons 2-13 in HSCs. Nevertheless, the study compares different templates and strategies to improve gene insertion. Readers may find these comparisons useful. However, these comparative studies do not include the DNA-PKcs inhibitor AZD7648 (PMID: 37580318, PMID: 37537500) which improves gene insertion more robustly than the methods presented here. Overall, the study presents a few interesting comparisons between different gene insertion templates and HR enhancing strategies. The results reinforce the current knowledge that the use of RNP+AAV results in the most efficient gene insertion and confirms challenges with the engraftment of edited HSCs. The significance of the results in advancing gene editing is unclear and is my reason for not recommending publication, at least in the current form. The comments below describe these concerns in more detail

Major comments:

The significance/novelty of the results presented in the manuscript (beyond the correction of CYBA mutations) is unclear and can be explained better. The standard in the field is the use of RNP and AAV to correct mutations or replace genes. The concern with that strategy is that it seems to limit the ability of HSCs to engraft in vivo. Nevertheless, it is not surprising that the use of RNP+AAV resulted in efficient gene correction of CYBA mutations. Similarly, the restoration of CYBB activity by insertion of exons 3-13 is consistent with previous studies. Although the study compares different strategies to improve gene insertion, these methods are established.

The discussion states that this strategy is the first to replace most of CYBB. This is an incorrect statement. A previous study in this journal compared different strategies to replace the CYBB cDNA. They showed that the replacement of exons 2-13 of

CYBB restores gp91phox expression and ROS production (PMID: 33712802). Does the approach presented here pose any advantages?

The reduction in the expansion of cells edited using ssODN and IDLV was more than the cells edited using AAV. Is this also mediated by P53 activation or is this mediated by a different mechanism? When considered with the fact that editing with IDLV and ssODN also reduced the ability of CD34+ to expand, can we attribute the reduced engraftment to AAV?

In Figure 4, the expansion of the edited CD34+ cells is comparable to Mock when editing is performed using HiFi Cas9. Do the authors think this might restore the ability of the HSCs to engraft in immunocompromised mice? Perhaps not, since other studies which show reduced engraftment potential also use high fidelity Cas9.

In terms of strategies to improve gene insertion, the small molecule AZD7648 is very effective in improving gene insertion compared to the strategies described here. Although I recognize that a comparison of editing with AZD7648 may be out of the scope of this current manuscript, it deserves to be at least discussed for completeness.

Minor comments:

Introduction does not discuss CYBA in much detail. Although they mention CYBA at the end when they describe the patient samples, it is easy to miss.

In Figure 2C they show that they can correct the mutant allele to a WT allele in a heterozygous sample. Coloring the increase in percent WT alleles differently might make the point clearer.

Reviewer #2

(Remarks to the Author)

The manuscript "Targeted gene editing and near universal cDNA insertion of CYBA and CYBB as a treatment for chronic granulomatous disease" submitted by Jonas H Wolff et al. describes two different gene editing strategies to treat chronic granulomatous disease. The first strategy aimed to correct c.282+1G>T in CYBA which causes p22phox-CGD using an HDR approach. After identifying the best performing sgRNA at this position, the authors used RNP electroporation plus rAAV6 for DNA donor template delivery for correction and achieved around 40% GFP reporter knock-in in healthy CD34 HSPCs. Unfortunately, the correction most likely induced a DNA damage response resulting in reduced proliferation, lower colony formation potential, poor engraftment in immunodeficient mice and loss of gene corrected cells in vivo. To overcome these limitations, the authors tested three different inhibitors/effectors to dampen the DNA damage response and increase HDR rates, namely i53, GSE56 and Ad5-E4orf6/7. The combination of all three inhibitors/effectors via mRNA co-delivery during electroporation significantly increased HDR rates in HSCs. Next, the authors tested which type of DNA donor template delivery (ssODN, IDLV or rAAV6) achieves the highest correction rate. Direct comparison revealed that rAAV6 outperformed the other two approaches resulting in higher viability and proliferation and achieving up to 70% HDR-mediated correction of the point mutation in one healthy donor. Upon granulocytic differentiation, the cells showed 40% functional NADPH oxidase activity based on ROS production. In the second part of the paper the authors targeted the c.252G>A mutation in CYBB, which causes gp91phox deficiency, the most common form of CGD. For this mutation, the authors used a similar HDR-based approach as for CYBA and corrected the point mutation in CD34+ HSPCs from a heterozygous carrier testing an allele-specific sgRNA targeting the mutated allele (sg2) and one that targeted both alleles (sg3). Using sg3, the authors achieved 50% HDR and restored ROS production to 30%. Next, instead of an individual correction approach, the authors tested a near universal correction approach by integrating a cDNA cassette containing exon 3-13 at the cut site of sg3. This near universal gene correction approach would be suited to treat around 86% of X-CGD patients. Using CD34+ HSPCs from the heterozygous carrier and the optimized HDR approach, the authors achieved 40% HDR-mediated correction in differentiated granulocytes based on GFP fluorescence, which resulted in over 50% ROS production. These cells were then transplanted into immunodeficient mice, where the authors observed a significant loss of engraftment in the sg3 treated cells. However, the few cells that did engraft maintained their HDR editing rate in vivo. The authors observed off-target editing of sg3 in multiple off-target sites, which was ameliorated by switching to HiFi Cas9 resulting in better viability, proliferation and colony formation.

The study is easy to follow for the reader, the data seems to be robust, and the conclusions are supported by the presented results. All graphs are well designed and easy to understand by themselves. Although the first two correction approaches are patient specific, the authors performed a comprehensive comparison between different inhibitors/effectors designed to dampen the DNA damage response and to increase HDR rates in CD34+ HSPCs and tested different DNA donor delivery methods, which could be interesting to a broad readership. In the discussion, the authors nicely discussed their findings addressing the controversy of AAV6 DNA donor delivery, discuss the three different HDR enhancers and the effect of the observed off-target editing. The method section is detailed and comprehensive and the supplements contain additional helpful information to reproduce the experiments.

The major shortcoming of this manuscript is the lack of editing in actual CGD patient-derived CD34+ HSPCs. It is known that due to chronic hyperinflammation in these patients, their HSCs are often exhausted and have impaired stem cell fitness, which reduces their transplantation capacity. Thus, it would be interesting to see if the presented editing approaches would allow correction of this very sensitive cell material and how toxic the combination of RNP electroporation plus rAAV6 transduction and the extended ex vivo cultivation periods would be on these cells. The authors showed that healthy cells already lose half of their engraftment potential after the correction procedure. Please at least discuss these aspects in the discussion section and how this could be mitigated in a translational approach.

The HDR approach using sg3, which targets both alleles, allows to study the overall correction rate and NADPH oxidase restoration as in the event of indel formation on the healthy allele, it gets disrupted (resembling a mutated allele) and only cells with proper HDR correction have functional NADPH oxidase activity. Unfortunately, this approach was impacted by the high off-target editing of sg3, which impaired the engraftment of edited CD34 cells. The authors then nicely showed that HiFi Cas9 could avoid off-target editing resulting in more colony formation as well as higher viability and proliferation. However, several key experiments are missing here. What is the HDR editing rate for the near universal approach in CYBB-HSPCs with sg3 HiFi Cas9? What is their ROS production? Upon transplantation of these cells into immunodeficient mice do they show improved engraftment and maintain their HDR editing rate?

In this regard, the authors should also study in more detail and in a more genome wide approach possible off-target effects of Cas9 and the rAAV6 donor. Are there any events of chromosomal rearrangements, translocations or chromotrypsis? Is there any integration of rAAV6 in the DNA cut site?

The authors designed a near universal gene correction approach which could treat 86% of X-CGD patients. Is there a specific reason, why the cDNA of exon 3-13 was integrated at the cut site in exon 3? Wouldn't it be more universal to add the full cDNA in exon 1 to treat potentially all X-CGD patients? Please explain in the main text the reasoning behind this choice.

There seems some discrepancy between HDR rate and ROS production in the different experiments. While in Figure 2 40% HDR results in 40% ROS production, the authors observed in Figure 3 first 30% ROS production with over 50% HDR correction and almost 60% ROS production with 40% HDR correction for the near universal approach. How do the authors explain these differences? How well does the correction rate correlates with NADPH oxidase activity? In this context what is the expression level of gp91phox after correction?

In the method section the authors describe that each experiment was started with 2 Mio CD34 HSPCs and the complete outgrowth was transplanted into immunodeficient mice. How many cells were transplanted per mouse? What was the cell loss during electroporation? Did the expansion culture help to increase cells numbers and to mitigate the loss during the gene correction procedure?

Overall, the time ex vivo seems quite long and could additionally impact the stemness of the CD34 HSPCs especially when treating actual CGD patient-derived HSPCs in the future. Would a shorter ex vivo time help while achieving similar HDR rates? Did the author test a shorter pre-stimulation of 24h or 48h?

Additional minor points are found in the following list:

Line 49: Correct to "NADPH oxidase".

Line 105: Correct to "using the most potent sgRNA (sg178) with an optimized rAAV6 at an MOI of 5000".

Line 149: Based on the supplement data, it looks like MOI 15,000 led to higher HDR rates and higher viability. Why wasn't this dose chosen for CD34 HSPCs?

Line 175: What are the therapeutically relevant levels for CGD gene correction? Please mention them here and maybe already in the introduction. This would help the reader to understand what the minimal goal in terms of gene correction would be.

Line 230: How were mock-treated cells treated? Were they electroporated with a control sgRNA? Or are they untreated as in the first figures?

Material and method section: Consider rearranging the paragraphs in chronologic order. Maybe move the "Viral vector production, purification and titration" section as well as the "In vitro transcription of mRNA" section in front of the paragraph about "Gene editing of cell lines and human CD34+ HSPCs".

Line 474 and 476: Correct to μL using the Greek letter.

Line 500: Which methylcellulose media was used? Please include the number of the medium or describe which cytokines were added to the medium.

Figure 4a: Please add statistical tests here.

Supplementary Figure S1: In the figure description correct to "8 days after nucleofection".

Supplementary Figure S4c: Correct to "Data represented as mean ($n=3$) \pm standard deviation". Additionally, please add a negative control of a FACS dot plot in the figure panel to show how the gate was set.

Supplementary Figure S6: Please correct the order in the figure description to rAAV6, ssODN and IDLV to match the order in the graphic.

Supplementary Figure S10a: Please correct the statistical comparisons to black lines above the data points.

Reviewer #3

(Remarks to the Author)

In this manuscript, the authors describe the optimization of CRISPR/Cas9-mediated gene correction strategies for two forms of chronic granulomatous disease involving CYBA and CYBB mutations. The methodology and analyses provided appear sound and the writing is clear. The authors compare different donors (single-stranded ODNs, integrase-defective lentivirus, and AAV6) for homology-directed repair and assess modulators of 53BP1 (i53) and p53 (Ad5-E4orf6/7 and GSE56) damage responses to improve editing efficiency and maintain hematopoietic stem cell viability and engraftment potential following editing. While many of these factors have been compared in prior studies, this study provides further data on the efficacy of these modulators in combination, particularly for enhancement of targeted insertion of AAV6 donors. The most notable finding appears to be the improvement of HSPC fitness by the high-fidelity HiFi Cas9 compared to regular-fidelity Cas9 for a guide RNA with a high degree of off-target activity (notably within two off-target genes, SSPOP and NOX1), although it is not clear whether these benefits on HSPC fitness would hold true with HiFi Cas9 for guide RNAs with lesser inherent off-target activities. However, the authors' claim that their CYBB cDNA insertion approach provides the most universal CYBB editing strategy described thus far in CD34+ HSPCs fails to consider a prior publication describing a more comprehensive CYBB cDNA insertion strategy tested in HSPCs, which limits the novelty and significance of the current study.

Major issues:

- 1) In the Discussion, the authors state "However, to our knowledge, the cDNA insertion strategy presented here, supporting targeted insertion of exon 3 through 13, is the closest to a universal X-CGD gene editing strategy yet reported in CD34+ HSPCs, potentially allowing treatment of more than 86% of X-CGD patients." A gene editing strategy involving targeted insertion of CYBB exon 2 through 13 has previously been tested in CD34+ HSPCs, as described by Sweeney et al., Correction of X-CGD patient HSPCs by targeted CYBB cDNA insertion using CRISPR/Cas9 with 53BP1 inhibition for enhanced homology-directed repair, *Gene Therapy* (2021). Since this prior approach included CYBB exon 2, it would allow for correction of a greater percentage of patients than the authors' exon 3 through 13 insertion approach.
- 2) It would be helpful to know whether the effects on HSPC fitness and engraftment with CYBB sg3 editing were due to the specific off-targets identified for sg3 (particularly SSPOP and NOX1 genes) or whether they were due to an overall reduction in off-target editing (including toxicity due to double-strand breaks and indels due to DNA repair without a template for HDR), which may indicate how generalizable these findings are for HiFi Cas9 usage to prevent HSPC damage during editing. If possible, an assessment of the effects of HiFi Cas9 on off-target editing by CYBB sg2 and on HSPC expansion, viability, and CFU activity after sg2 editing with HiFi versus regular Cas9 might indicate whether HiFi Cas9 can further improve editing and HSPC fitness with other guide RNAs that lack the high off-target activity displayed by CYBB sg3.
- 3) One of the readouts for assessing the degree of correction in prior CYBB editing strategies and retroviral or lentiviral gene therapy trials has been the level of gp91phox/NOX2 protein expression or ROS production achieved per corrected cell (such as by mean fluorescence intensity measurement of positive gated cells by flow cytometry analysis). For your AAV correction data in Figure 2g, how does the MFI of the gated ROS+ population compare with the MFI of the gated ROS+ population of Mock-treated cells?
- 4) The authors acknowledged reports of Cas9-nuclease induced double strand DNA breaks and risks of genome instability. Chromosomal rearrangement, translocations, or chromothripsis are important risks of DSBs that need to be assessed for by evaluating for large chromosomal structural variants.
- 5) The evaluation for off-targets based on in silico CRISPRoff nominated sites is too limited. A guide independent assessment is important for safety consideration for clinical application.
- 6) A major drawback with AAV donors is the risks of concatemers which the authors discussed but did not assess for. The standard amplicon sequencing used will not differentiate between a single copy insert or concatemers, thereby an accurate assessment for concatemers has implications for actual rates of functional gene correction with a single copy insert.

Minor issues:

- HiFi Cas9 can be associated with varying loss of on-target efficiency at some target sites compared to regular Cas9 (~25% reduction in on-target efficiency across the target sites assessed by Vakulskas et al., 2018). Based on the data in Fig. 4e, the authors might highlight that the on-target editing efficiency for CYBB sg3 with HiFi Cas9 was nearly the same as regular Cas9, but that a more substantial loss of on-target efficiency could occur with other guide RNAs.
- Some portion of patients with X-CGD have large deletions of X-chromosome regions encompassing CYBB and neighboring genes (i.e. McLeod syndrome), which would not be correctable by any editing strategy targeting the CYBB gene. Does the data shown in Figure 3g and Supplementary Figure S9 include those patients? If not, you should clarify that your calculation of >82% correctable is for X-CGD patients without large X-chromosome deletions encompassing the entire CYBB gene.

Overall, the studies are well designed. There remains a lot of work before these approaches can be applied to clinical translation, if at all, since evaluation of the safety aspects is insufficient for consideration for human application.

A major flaw is not designing studies to address current and known concerns regarding the use of Cas9 nuclease or AAV-related adverse events. Targeted insertion of CYBB 'gene' and use of agents to address inefficient HDR and DDR is not novel.

Reviewer #4

(Remarks to the Author)

Version 1:

Reviewer comments:

Reviewer #1

(Remarks to the Author)

In this revised manuscript, the authors have addressed many of the previous comments. They have included additional experiments reporting off-target editing profile in CD34+ cells edited using WT, HiFi Cas9 and paired nickase editing strategy. They have characterized engraftment of CD34+ cells edited using HiFi Cas9 in mice and also characterize P21 activation in edited CD34+ cells.

However, my main concern about the significance and novelty of these findings have not been addressed. For CYBB gene correction, Sweeney et al (Gene therapy, 2021) have previously already demonstrated the replacement of the CYBB exons 2-13 in CD34+ cells from people with X-CGD (which is absent in Wolff et al). Sweeney et al already demonstrated engraftment comparable to healthy control CD34+ cells and reported no significant off-target edits. Furthermore, Sweeney et al already showed restoration of ROS function in cells in vitro and in vivo after engraftment using corrected CD34 cells obtained from a person with X-CGD. Wolff et al only report editing in CD34+ cells from a heterozygous healthy individual and report ROS function in those cells.

Apart from gene correction, Wolff et al report additional conditions that may improve the safety of a CYBB gene therapy. However, these strategies are already the standard in the field. It is widely known in the field that the use of high fidelity Cas9 (R691A) is best for reducing off-target activity (PMID: 30082871). The use of paired nickases to reduce off-target activity (PMID: 35658035) and translocations has also been reported before (PMID: 38459694). In my assessment, the study reports a system to replace a smaller section of CYBB (exons 3-13 as opposed to exons 2-13), does not show restored ROS function in corrected cells and uses reagents more prone to off-target activity compared to previous work. The solutions to reduce aberrant genetic changes are all already known and widely reported. Therefore, the significance is really not clear to me.

Similarly, the first part of the manuscript describes the correction of an individual mutation using ssODN, AAV and IDLV. However, the findings are predictable. It is well-established in the field that AAV6-based gene correction enables efficient gene insertion (PMID: 27820943) and is superior compared to ssODN, IDLV and adenovirus vectors (PMID: 31178391). Thus, other than the application of the tools to a new disease setting, these findings also do not seem novel to me.

Additional presentation comments include the following:

1. For Figure 4, it is not clear from the results section and figure captions that the ROS function was measured in CD34+ cells from a heterozygous or unaffected individuals. This is especially the case with the section describing the exon 3-13 cDNA
2. The CAST-seq results will be more useful to readers if they report the percent of alleles with aberrant editing.
3. The discussion mentions potential loss of activity while using High fidelity Cas9. However, there are a few different high fidelity variants. This variant (R691A) used in this study is specifically one that maintains on-target activity comparable to WT-Cas9 while reducing off-target activity.

Reviewer #2

(Remarks to the Author)

In the revised manuscript "Targeted gene editing and near universal cDNA insertion of CYBA and CYBB as a treatment for chronic granulomatous disease", the authors diligently addressed all comments and suggestions of this reviewer. The authors added additional experiments to the manuscript, which further strengthened the findings of the study and added additional value to the manuscript. The revised manuscript will be of great interest to a broad readership, especially in the gene therapy and CGD field.

Reviewer #3

(Remarks to the Author)

Line 64: "The best treatment for CGD...". It might be better to spell out what allogeneic transplant can/can't do since the term 'best' implies there are others to compare with, and it has problems as the authors described.

Line 182: It is not strictly accurate to say that 20% ROS+ cells is a threshold for reconstituting neutrophil function. Rather, 20% ROS+ neutrophils in X-CGD patients provides a level of neutrophil function that will likely be sufficient to provide protection from CGD-typical infections.

Lines 199-203: This is somewhat awkwardly phrased and makes it a little unclear that the sgRNAs were tested separately. Consider changing “We therefore moved on with sg2 and sg3. Using these sgRNAs...” to something like “We therefore focused on assessing sg2 versus sg3 in combination with a rAAV6 repair template designed to correct the CYBB c.252G>A variant and install silent mutations to disrupt the binding sites of all sgRNA. This resulted in up to 20% HDR in CYBB c.252G>A heterozygous PBMCs for the sg3 sgRNA that targets CYBB without allele specificity (Fig. 3b). However, limited HDR was observed using the allele-specific sg2.”

Line 239: Please include a statement regarding the limitations of using ddPCR for assessment of intact, targeted integration, since it cannot itself determine the integrity of insert sequences but rather relies on insert-specific primers/probe for copy number analysis (which could include inserts with partial missing or altered sequences).

Lines 303-309: Please include specific values/numbers/ranges in the text for engraftment rates, percentage GFP+, and gene editing levels rather than using descriptive terms such as ‘substantial loss of gene editing levels’, ‘engrafted comparably’, ‘highest GFP expression’.

Line 335-337: The authors state that “In contrast, using D10A Cas9n did not significantly lower CFU potential (Fig. 5e, Supplementary Fig. S14b), indicating very limited cytotoxicity of our D10A Cas9n gene editing strategy.” It would be more accurate to state that “In contrast, using D10A Cas9n did not significantly lower CFU frequency...”, since although D10A Cas9n did not significantly reduce the number of CFUs (Fig. 5e), it did appear to have significantly reduced the total number of cells per plate in the CFU assay compared to mock-treated (Supplementary Fig. S14b; *p<0.05 for D10A Cas9n versus mock-treated), suggesting that CFU sizes may have been reduced by D10A Cas9n treatment. This might be regarded as decreasing “CFU potential” in terms of the overall proliferative potential of the cells within the colonies, although not in terms of the frequency of colony-forming units present in the treated HSPC population.

Reviewer #4

(Remarks to the Author)

NCOMMS-24-40917

Dear Reviewers,

Thank you for your careful review of our manuscript entitled 'Targeted gene editing and near-universal cDNA insertion of CYBA and CYBB as a treatment for chronic granulomatous disease'. We appreciate the positive and constructive feedback. We have revised the manuscript accordingly based on extensive additional experimentation as requested, which has added new important findings and substantial additional novelty to the manuscript.

Most notably, we did the following:

- We expanded and improved off-target DNA cleavage analysis by performing DISCOVER-seq to further nominate potential off-target sites. Additionally, these sites have been validated by high-throughput sequencing. This analysis further strengthens our demonstration of differences between standard and HiFi Cas9, although off-target cleavage is also detected with HiFi Cas9.
- To further support this finding, we also carried out new engraftments studies in mice to show that the use of HiFi Cas9 rescues the loss of engraftment potential previously observed.
- To address the safety issues that we unveil for both standard and HiFi Cas9 we set out to develop a novel gene editing approach for X-CGD that employs paired Cas9 D10A nickases. We show that this strategy retains high on-target efficacy with no detectable off-target effects.
- To further compare the safety profiles of all HDR-based approaches (standard Cas9, HiFi Cas9, and Cas9 D10A nickases), we performed CAST-seq on stem cells engineered with all three gene editing approaches demonstrating marked differences in the formation of off-target mediated translocations. Importantly, we do not detect any translocations using the newly developed Cas9 D10A nickase strategy, highlighting the advance of this work in relation to previously published therapeutic gene editing strategies for X-CGD.

In summary, we believe that the work presented here delivers new insight into how to address safety and off-target editing related to gene editing of *CYBB* as a treatment option for X-CGD. We highlight key challenges that these strategies face, and report on a novel gene editing approach in X-CGD that offers a solution to these challenges.

We re-submit a revised version of the manuscript with all changes indicated in red. Please find below a point-by-point response to all comments with our responses marked in red font.

Reviewer #1:

Major comments:

The significance/novelty of the results presented in the manuscript (beyond the correction of CYBA mutations) is unclear and can be explained better. The standard in the field is the use of RNP and AAV to correct mutations or replace genes. The concern with that strategy is that it seems to limit the ability of HSCs to engraft in vivo.

Nevertheless, it is not surprising that the use of RNP+AAV resulted in efficient gene correction of CYBA mutations. Similarly, the restoration of CYBB activity by insertion of exons 3-13 is consistent with previous studies. Although the study compares different strategies to improve gene insertion, these methods are established.

Response: We acknowledge the reviewer's critical view on this issue. However, we believe that the gene editing of CYBA, as well as the characterizations and comparisons of different repair templates and Cas9 variants in relation to CYBB, offers new insights into key challenges of both this work and previously published gene editing strategies targeting CYBB. Some of the key points that were made in the original manuscript have been strengthened now with additional data on off-target profiles (using DISCOVER-seq) and formation of translocations (using CAST-seq). To further address the reviewer's point, adding further novelty to this work, we have also performed new engraftment studies to show that engraftment potential is rescued by lowering the off-target profile using HiFi Cas9. Notably, considering the detection of off-target editing with both above-mentioned techniques (standard and HiFi Cas9), we also established a new paired Cas9 D10A nickase strategy. We report that this approach does not result in detectable off-target effects, thus offering a solution of previously identified challenges of CYBB gene editing. Together, we believe that these new additions to the manuscript add significant insights into gene editing of not only CYBA and CYBB, but to RNP+AAV-mediated gene editing of CD34+ HSPCs in general.

The discussion states that this strategy is the first to replace most of CYBB. This is an incorrect statement. A previous study in this journal compared different strategies to replace the CYBB cDNA. They showed that the replacement of exons 2-13 of CYBB restores g91phox expression and ROS production (PMID: 33712802). Does the approach presented here pose any advantages?

Response: We agree that we should have given these findings more attention and acknowledge that the statement that we included in the discussion was not correct. We have rephrased the wording in the discussion accordingly (line 466-468) and included this and an additional reference (refs 65 and 66) where relevant in the discussion. Our mission was to work through different HDR-based approaches for correction in CGD/X-CGD and describe targeting profiles and genomic integrity for each approach, allowing us to point to key challenges and strategies for solving such challenges. For the cDNA insertion approach, we used the same sgRNA (sg3) that was used for targeted repair and thus with the same off-target cleavage profile. Based on the characterization of off-target editing, we show improved safety with a cDNA insertion approach based on using HiFiCas9, supporting engraftment, however still with a documented risk of off-target cleavage. To address this issue, we moved to a paired nickase approach, demonstrating further reduction of off-target cleavage (Figure 5f), also compared to sg3 with HiFiCas9. Assuming that a cDNA strategy is compatible with a paired D10A Cas9n approach, such strategy offers significant improvements regarding off-target editing and off-target mediated translocations.

The reduction in the expansion of cells edited using ssODN and IDLV was more than the cells edited using AAV. Is this also mediated by P53 activation or is this mediated by a different mechanism? When considered with the fact that editing with IDLV and

ssODN also reduced the ability of CD34+ to expand, can we attribute the reduced engraftment to AAV?

Response: This is another excellent comment. Although we do not directly assay the impact of ssODNs, IDLV and rAAV6 on p53-activation and engraftment potential, others have previously shown that these modalities do themselves trigger p53-activation and reduce engraftment potential (PMID: 36206730, PMID: 30905619). Therefore, based on previous reports, we believe that the reduction in engraftment potential stems from both the AAV and the Cas9-induced DSBs. To address this question further, we have performed p21 qPCR and included new data to show reduced p53-activation following the use of HiFi Cas9 and D10A Cas9n to limit off-target DSBs (Figure 5d).

In Figure 4, the expansion of the edited CD34+ cells is comparable to Mock when editing is performed using HiFi Cas9. Do the authors think this might restore the ability of the HSCs to engraft in immunocompromised mice? Perhaps not, since other studies which show reduced engraftment potential also use high fidelity Cas9.

Response: We thank the reviewer for valuable feedback, this is another relevant point. We have carried out additional engraftment experiments to investigate this exact question. Notably, we found that the use of HiFi Cas9 rescued the engraftment potential of gene-edited CYBB-HSPCs to a level comparable to mock-treated HSPCs (Figure 4d). We believe that this demonstrates that reduction in engraftment potential is highly dependent on overall DSB formation and that the engraftment potential may thus be directly negatively affected by off-target editing to a much higher degree than previously reported.

In terms of strategies to improve gene insertion, the small molecule AZD7648 is very effective in improving gene insertion compared to the strategies described here. Although I recognize that a comparison of editing with AZD7648 may be out of the scope of this current manuscript, it deserves to be at least discussed for completeness.

Response: Regarding AZD7648, we acknowledge that the inhibitor has been shown to robustly increase gene insertion rates. Nevertheless, we did not include the inhibitor as we found it to have detrimental effects in both cell lines and HSPCs in the early phase of this work. This has recently been supported by findings published by the Corn lab, showing that AZD7648 causes excessive large deletions at the on-target site (PMID: 39604565). We have added a small paragraph in the discussion addressing this point (line 409-412).

Minor comments:

Introduction does not discuss CYBA in much detail. Although they mention CYBA at the end when they describe the patient samples, it is easy to miss.

Response: We have revised the introduction to add more information related to CYBA (line 62-64).

In Figure 2C they show that they can correct the mutant allele to a WT allele in a

heterozygous sample. Coloring the increase in percent WT alleles differently might make the point clearer.

Response: We understand the suggestion, but, as we cannot distinguish between true wild-type (i.e. unedited wild-type) and edited wild-type (i.e. mutant to wt conversion), we believe that the fairest representation is the one shown.

Reviewer #2:

Major comments:

The major shortcoming of this manuscript is the lack of editing in actual CGD patient-derived CD34+ HSPCs. It is known that due to chronic hyperinflammation in these patients, their HSCs are often exhausted and have impaired stem cell fitness, which reduces their transplantation capacity. Thus, it would be interesting to see if the presented editing approaches would allow correction of this very sensitive cell material and how toxic the combination of RNP electroporation plus rAAV6 transduction and the extended ex vivo cultivation periods would be on these cells. The authors showed that healthy cells already lose half of their engraftment potential after the correction procedure. Please at least discuss these aspects in the discussion section and how this could be mitigated in a translational approach.

Response: We thank the reviewer for comments and constructive criticism. We do indeed agree that inclusion of HSPCs from a CGD patient would have been both relevant and eventually crucial for clinical translation. However, due to Danish regulations regarding paediatric patients, it was not possible for us to acquire HSPCs from patients under the age of 18. We have included a discussion of these aspects and the limitations that this issue represents (line 453-459). In relation to the loss of engraftment potential, we have carried out additional engraftment experiments which show that HiFi Cas9 rescues the engraftment potential of gene edited CYBB-HSPCs to a level comparable to mock-treated HSPCs (Figure 4d).

The HDR approach using sg3, which targets both alleles, allows to study the overall correction rate and NADPH oxidase restoration as in the event of indel formation on the healthy allele, it gets disrupted (resembling a mutated allele) and only cells with proper HDR correction have functional NADPH oxidase activity. Unfortunately, this approach was impacted by the high off-target editing of sg3, which impaired the engraftment of edited CD34 cells. The authors then nicely showed that HiFi Cas9 could avoid off-target editing resulting in more colony formation as well as higher viability and proliferation. However, several key experiments are missing here. What is the HDR editing rate for the near universal approach in CYBB-HSPCs with sg3 HiFi Cas9? What is their ROS production? Upon transplantation of these cells into immunodeficient mice do they show improved engraftment and maintain their HDR editing rate?

Response: This is certainly valid criticism which is in line with similar concerns that were raised by reviewer #1. As described above in our response to reviewer #1's comments, we have conducted additional engraftment studies using HiFi Cas9-treated HSPCs. This also includes a comparison of HDR rates using WT and HiFi Cas9 both before and after transplantation in mice (Figure 4f). Crucially, this shows that the loss

in engraftment potential is rescued by employing HiFi Cas9 and leads to a better preservation of gene-edited cells in transplanted HSPCs.

In this regard, the authors should also study in more detail and in a more genome wide approach possible off-target effects of Cas9 and the rAAV6 donor. Are there any events of chromosomal rearrangements, translocations or chromotrypsis? Is there any integration of rAAV6 in the DNA cut site?

Response: Yes, we agree that further analyses are relevant. We have carried out additional off-target analyses. More specifically, we performed DISCOVER-seq of both standard Cas9 and HiFi Cas9-treated cells to better nominate potential off-target sites (Figure 4a-b). We also investigated the insertion of AAV ITR sequences at the on-target sites and added additional data (Supplementary Fig. S14a). Moreover, we performed CAST-seq to identify possible translocations (Figure 4g-h). Importantly, although fewer off-targets were detected with HiFi Cas9, we did indeed detect translocations using HiFi Cas9. Therefore, we moved on to develop a paired D10A Cas9 nickase gene editing strategy, which was found to retain high HDR rates but did not lead to detectable off-target editing and did not cause detectable translocations. Overall, in the revised manuscript, a panel of editing strategies and their engraftment capacities are presented in relation to genome-wide assessments of off-target cleavage and formation of translocations, demonstrating a path toward a clinically relevant correction strategy for a specific target variant.

The authors designed a near universal gene correction approach which could treat 86% of X-CGD patients. Is there a specific reason, why the cDNA of exon 3-13 was integrated at the cut site in exon 3? Wouldn't it be more universal to add the full cDNA in exon 1 to treat potentially all X-CGD patients? Please explain in the main text the reasoning behind this choice.

Response: When developing gene insertion strategies, it is generally recognized that retaining intron 1 is required for proper expression. For CYBB it has previously been established that intron 1 is indeed required for correct expression (PMID: 28153086, PMID: 33712802). Additionally, by inserting from exon 3, we would also be able to use the same sgRNAs for both cDNA insertion and direct target gene correction. We therefore chose to insert the partial CYBB cDNA starting from exon 3.

There seems some discrepancy between HDR rate and ROS production in the different experiments. While in Figure 2 40% HDR results in 40% ROS production, the authors observed in Figure 3 first 30% ROS production with over 50% HDR correction and almost 60% ROS production with 40% HDR correction for the near universal approach. How do the authors explain these differences? How well does the correction rate correlates with NADPH oxidase activity? In this context what is the expression level of gp91phox after correction?

Response: We acknowledge the confusion these data may cause. The discrepancy between HDR rates and ROS production is due to variations in the efficiency of the myeloid differentiation. The differentiation to neutrophil granulocytes using the methods described did not result in complete differentiation to neutrophil granulocytes, as demonstrated by only achieving between 40 and 60% CD15+ myeloid cells following 14 days of differentiation (Figure 3d). Furthermore, CD15+ is not specific to

only neutrophil granulocytes, which are the cell of interest for the oxidative burst assay. As such, the total ROS+ cells should for each assay be viewed in relation to the mock control. Accordingly, we have changed the way the oxidative burst assays are represented, so that they display the ROS+ cells in relation to the mock control (Figure 3e and 3j). Non-normalized plots are available to readers in Supplementary Fig. S8.

In the method section the authors describe that each experiment was started with 2 Mio CD34 HSPCs and the complete outgrowth was transplanted into immunodeficient mice. How many cells were transplanted per mouse? What was the cell loss during electroporation? Did the expansion culture help to increase cells numbers and to mitigate the loss during the gene correction procedure?

Response: Point taken. We have included tables in Supplementary Table S7 and S8 with stats regarding the setup of engraftment experiments, including number of cells treated, cells after expansion, fold expansion, viability after expansion, resuspension volumes, injected volumes, and total cells injected.

Overall, the time ex vivo seems quite long and could additionally impact the stemness of the CD34 HSPCs especially when treating actual CGD patient-derived HSPCs in the future. Would a shorter ex vivo time help while achieving similar HDR rates? Did the author test a shorter pre-stimulation of 24h or 48h?

Response: We used an expansion time of 72h based on established work by others in the field (PMID: 24870228) demonstrating that, when HSPCs are cultured using SR1, nucleofection of the cells after 72h expansion results in highest number of cells with targeted integration following engraftment into immunodeficient mice. We do recognize that the culture time is a possible area that could be optimized, but did not test this further in this work. We repeatedly found that mock-treated HSPCs engraft very well following this expansion protocol, and so we believe that this indicates that the stemness of the HSPCs is not severely affected by this expansion. Still, although we deemed such investigations to be out of scope for this work, we acknowledge that this would be interesting to investigate in the future.

Minor comments:

Line 49: Correct to “NADPH oxidase”.

Response: Yes, corrected.

Line 105: Correct to “using the most potent sgRNA (sg178) with an optimized rAAV6 at an MOI of 5000”.

Response: This is now corrected.

Line 149: Based on the supplement data, it looks like MOI 15,000 led to higher HDR rates and higher viability. Why wasn't this dose chosen for CD34 HSPCs?

Response: Based on the known detrimental effect of AAV on HSPC fitness, we always go with the lowest possible MOI to avoid cell loss following gene editing.

Line 175: What are the therapeutically relevant levels for CGD gene correction?

Please mention them here and maybe already in the introduction. This would help the reader to understand what the minimal goal in terms of gene correction would be.

Response: Agree, this is a valid point. We have added additional text concerning the expected therapeutic level (line 182-184).

Line 230: How were mock-treated cells treated? Were they electroporated with a control sgRNA? Or are they untreated as in the first figures?

Response: We have added this information in the methods section (line 569-570).

Material and method section: Consider rearranging the paragraphs in chronological order. Maybe move the “Viral vector production, purification and titration” section as well as the “In vitro transcription of mRNA” section in front of the paragraph about “Gene editing of cell lines and human CD34+ HSPCs”.

Response: We have reordered the methods sections as suggested by the reviewer.

Line 474 and 476: Correct to μL using the Greek letter.

Response: Corrected.

Line 500: Which methylcellulose media was used? Please include the number of the medium or describe which cytokines were added to the medium.

Response: The medium used is now described in the methods section (line 614)

Figure 4a: Please add statistical tests here.

Response: Added. This particular figure is now Supplementary Fig. S10b

Supplementary Figure S1: In the figure description correct to “8 days after nucleofection”.

Response: ‘6 days after nucleofection’ is correct.

Supplementary Figure S4c: Correct to “Data represented as mean ($n=3$) \pm standard deviation”. Additionally, please add a negative control of a FACS dot plot in the figure panel to show how the gate was set.

Response: This is now corrected. Unstained and biological control samples are now shown (Supplementary Fig. S4d).

Supplementary Figure S6: Please correct the order in the figure description to rAAV6, ssODN and IDLV to match the order in the graphic.

Response: Corrected.

Supplementary Figure S10a: Please correct the statistical comparisons to black lines above the data points.

Response: Corrected.

Reviewer #3 (Remarks to the Author):

Major comments:

1) *In the Discussion, the authors state “However, to our knowledge, the cDNA insertion strategy presented here, supporting targeted insertion of exon 3 through 13, is the closest to a universal X-CGD gene editing strategy yet reported in CD34+ HSPCs, potentially allowing treatment of more than 86% of X-CGD patients.” A gene editing strategy involving targeted insertion of CYBB exon 2 through 13 has previously been tested in CD34+ HSPCs, as described by Sweeney et al., Correction of X-CGD patient HSPCs by targeted CYBB cDNA insertion using CRISPR/Cas9 with 53BP1 inhibition for enhanced homology-directed repair, Gene Therapy (2021). Since this prior approach included CYBB exon 2, it would allow for correction of a greater percentage of patients than the authors’ exon 3 through 13 insertion approach.*

Response: We acknowledge the reviewer’s critique, which overlaps with comments from both reviewer #1 and #2. As noted in the responses above, we do agree that these previous reports should be acknowledged and included. Accordingly, we have revised parts of the discussion to better reflect our findings in relation to previously published work. Please also see comments to reviewer #1 (response #2) and reviewer #2 (response #4).

2) *It would be helpful to know whether the effects on HSPC fitness and engraftment with CYBB sg3 editing were due to the specific off-targets identified for sg3 (particularly SSPOP and NOX1 genes) or whether they were due to an overall reduction in off-target editing (including toxicity due to double-strand breaks and indels due to DNA repair without a template for HDR), which may indicate how generalizable these findings are for HiFi Cas9 usage to prevent HSPC damage during editing. If possible, an assessment of the effects of HiFi Cas9 on off-target editing by CYBB sg2 and on HSPC expansion, viability, and CFU activity after sg2 editing with HiFi versus regular Cas9 might indicate whether HiFi Cas9 can further improve editing and HSPC fitness with other guide RNAs that lack the high off-target activity displayed by CYBB sg3.*

Response: We thank the reviewer for this comment. None of the identified off-targets had reported critical functions in HSPCs and therefore we hypothesized the cytotoxicity could be attributed to excess DSB formation. To further support this argument, we performed qPCR to assay p53-activation, and indeed found that use of standard Cas9 significantly increased p21 expression, indicating increased p53-activation and DDR (Figure 5d). Additionally, we conducted new engraftment studies to show that use of HiFi Cas9 also rescued the observed loss of engraftment potential (Figure 4d). Although we agree that an investigation of the effect of HiFi Cas9 with sg2 would be interesting, we instead chose to focus on lowering the off-target effects documented for sg3. We thought this was more relevant, as sg2 is patient-specific and thus not the main sgRNA of interest for developing this gene editing strategy.

3) *One of the readouts for assessing the degree of correction in prior CYBB editing strategies and retroviral or lentiviral gene therapy trials has been the level of gp91phox/NOX2 protein expression or ROS production achieved per corrected cell (such as by mean fluorescence intensity measurement of positive gated cells by flow cytometry analysis). For your AAV correction data in Figure 2g, how does the MFI of the gated ROS+ population compare with the MFI of the gated ROS+ population of Mock-treated cells?*

Response: We have included plots showing MFI values of ROS+ cells in Supplementary Figure S8.

4) The authors acknowledged reports of Cas9-nuclease induced double strand DNA breaks and risks of genome instability. Chromosomal rearrangement, translocations, or chromothrypsis are important risks of DSBs that need to be assessed for by evaluating for large chromosomal structural variants.

Response: We fully recognize this point and agree that investigation of large chromosomal aberrations is a crucial for gene editing therapies like the one reported here and for the major points we make in the manuscript. To address this point, we performed CAST-seq to identify off-target mediated translocations (OMTs) (Figure 5g-h). Notably, we report that HiFi Cas9 reduces OMTs compared to standard Cas9, although it does not remove OMTs. We also carried out extensive experimental work to develop a paired Cas9 D10A nickase strategy, which is included in the revised manuscript. Notably, CAST-seq did not identify OMTs for the nickase-based approach, demonstrating important safety differences between the different editing strategies.

5) The evaluation for off-targets based on in silico CRISPRoff nominated sites is too limited. A guide independent assessment is important for safety consideration for clinical application.

Response: Yes, we acknowledge that the off-target nomination based solely on in silico prediction was too limited. In accordance, we performed DISCOVER-seq of sg3 to further nominate potential off-targets. We performed DISCOVER-seq on cells treated with either standard Cas9 or HiFi Cas9 and included any hits found in the off-target analysis. Data are included in a new Figure 4 (panels 4a and 4b).

6) A major drawback with AAV donors is the risks of concatemers which the authors discussed but did not assess for. The standard amplicon sequencing used will not differentiate between a single copy insert or concatemers, thereby an accurate assessment for concatemers has implications for actual rates of functional gene correction with a single copy insert.

Response: We thank the reviewer for this comment. Similar criticism was raised by reviewer #2. We are aware of recent work showing integration of AAV concatemers (PMID 38589662), which we also elaborate on in the discussion (line 380-384). The recent study by Suchy et al. also demonstrated the difficulty in assaying these concatemeric insertions due to inability of PCR-based methods to detect them. As such, we believe that an accurate and comprehensive characterization of these concatemers is beyond the scope of the present work. Nevertheless, as stated in the discussion, we do acknowledge that such investigations are needed in the future before moving on to clinical translation. However, we have assayed the insertion of AAV ITR sequences at the on-target sites (Supplementary Fig. S14a). In relation to the implications for functional gene correction, we do acknowledge that these concatemeric insertions can lead to discrepancy between genomic correction rates and functional rescue. Despite of this, we would like to highlight that we are able to

show functional correction in terms of percentage ROS+ cells, directly demonstrating the level of functional gene correction.

Minor issues:

- *HiFi Cas9 can be associated with varying loss of on-target efficiency at some target sites compared to regular Cas9 (~25% reduction in on-target efficiency across the target sites assessed by Vakulskas et al., 2018). Based on the data in Fig. 4e, the authors might highlight that the on-target editing efficiency for CYBB sg3 with HiFi Cas9 was nearly the same as regular Cas9, but that a more substantial loss of on-target efficiency could occur with other guide RNAs.*

Response: Yes, we agree. We have highlighted this finding as suggested (line 275) and noted that similar findings might be relevant for other sgRNAs (line 427-432).

- *Some portion of patients with X-CGD have large deletions of X-chromosome regions encompassing CYBB and neighboring genes (i.e. McLeod syndrome), which would not be correctable by any editing strategy targeting the CYBB gene. Does the data shown in Figure 3g and Supplementary Figure S9 include those patients? If not, you should clarify that your calculation of >82% correctable is for X-CGD patients without large X-chromosome deletions encompassing the entire CYBB gene.*

Response: This is another valid point. We have clarified how the correctable variants have been calculated (line 840-842).

Overall, the studies are well designed. There remains a lot of work before these approaches can be applied to clinical translation, if at all, since evaluation of the safety aspects is insufficient for consideration for human application. A major flaw is not designing studies to address current and known concerns regarding the use of Cas9 nuclease or AAV-related adverse events. Targeted insertion of CYBB 'gene' and use of agents to address inefficient HDR and DDR is not novel.

Response: We appreciate the reviewer's feedback. Similar criticism was posed by reviewer #1 (response #1). Overall, we believe that we have used these comments to improve the manuscript further. Please see the response to reviewer #1. In terms of not addressing safety concerns of RNP+AAV-based gene editing, we have added substantial new experimental work to address this. We have expanded off-target profiling by adding both DISCOVER-seq data and CAST-seq analyses. We have also performed new engraftment studies to show that a lower off-target profile can rescue engraftment potential. Moreover, we have established a new paired Cas9 D10A nickase strategy that does not result in detectable off-target effects, thus offering a solution of previously identified challenges of CYBB gene editing. Together, we believe that these new additions to the manuscript add significant insights into gene editing of not only CYBA and CYBB, but to RNP+AAV-mediated gene editing of CD34+ HSPCs in general.

Reviewer #4:

I co-reviewed this manuscript with one of the reviewers who provided the listed reports.

Response: We thank reviewer #4 for her/his contributions to the review of this manuscript.

Thank you so much again for your constructive feedback.

Sincerely,
Jacob Giehm Mikkelsen, Professor, PhD
Corresponding author

Dear reviewers,

We wish to thank you again for your effort and time providing constructive criticism of the revised version of our manuscript entitled 'Targeted gene editing and near-universal cDNA insertion of CYBA and CYBB as a treatment for chronic granulomatous disease'. In the new version of the manuscript, we have addressed all comments and modified as suggested.

We re-submit a revised version of the manuscript with all changes indicated in red. Please find below a point-by-point response to all comments with our responses marked in red font.

Reviewer #1:

For Figure 4, it is not clear from the results section and figure captions that the ROS function was measured in CD34+ cells from a heterozygous or unaffected individuals. This is especially the case with the section describing the exon 3-13 cDNA

Response: Yes, we agree. We have updated the figure legends to clarify which cells were used for the oxidative burst assay.

The CAST-seq results will be more useful to readers if they report the percent of alleles with aberrant editing.

Response: We agree that these data are important for the interpretation of the CAST-seq results. We have included data showing on-target efficacy of cells used for CAST-seq analysis in Supplementary Fig. 15a.

The discussion mentions potential loss of activity while using High fidelity Cas9. However, there are a few different high fidelity variants. This variant (R691A) used in this study is specifically one that maintains on-target activity comparable to WT-Cas9 while reducing off-target activity.

Response: Yes, this is a fair point. We have rephrased the paragraph in question to clarify this fact (line 423-426).

Reviewer #2:

In the revised manuscript "Targeted gene editing and near universal cDNA insertion of CYBA and CYBB as a treatment for chronic granulomatous disease", the authors diligently addressed all comments and suggestions of this reviewer. The authors added additional experiments to the manuscript, which further strengthened the findings of the study and added additional value to the manuscript. The revised manuscript will be of great interest to a broad readership, especially in the gene therapy and CGD field.

Response: We appreciate the positive feedback of reviewer #2 and we are pleased to hear that our revisions have been satisfactory.

Reviewer #3:

Line 64: “The best treatment for CGD...”. It might be better to spell out what allogeneic transplant can/can’t do since the term ‘best’ implies there are others to compare with, and it has problems as the authors described.

Response: Yes, we agree. We have changed the wording accordingly. The sentence now reads: ‘The current treatment for CGD relies on allogeneic hematopoietic stem cell (HSC) transplantation....’ (line 54-55).

Line 182: It is not strictly accurate to say that 20% ROS+ cells is a threshold for reconstituting neutrophil function. Rather, 20% ROS+ neutrophils in X-CGD patients provides a level of neutrophil function that will likely be sufficient to provide protection from CGD-typical infections.

Response: Good point, thanks for clarifying this. We have modified the sentence and utilized almost the same wording as the reviewer (line 171-174).

Lines 199-203: This is somewhat awkwardly phrased and makes it a little unclear that the sgRNAs were tested separately. Consider changing “We therefore moved on with sg2 and sg3. Using these sgRNAs...” to something like “We therefore focused on assessing sg2 versus sg3 in combination with a rAAV6 repair template designed to correct the CYBB c.252G>A variant and install silent mutations to disrupt the binding sites of all sgRNA. This resulted in up to 20% HDR in CYBB c.252G>A heterozygous PBMCs for the sg3 sgRNA that targets CYBB without allele specificity (Fig. 3b). However, limited HDR was observed using the allele-specific sg2.”

Response: We agree with this suggestion and have inserted the text that was suggested by the reviewer (line 190-195).

Line 239: Please include a statement regarding the limitations of using ddPCR for assessment of intact, targeted integration, since it cannot itself determine the integrity of insert sequences but rather relies on insert-specific primers/probe for copy number analysis (which could include inserts with partial missing or altered sequences).

Response: We agree that the ddPCR assay will not comprehensively discriminate between intact and partial cDNA integrations. We chose to include a statement related to this issue in the discussion (line 449-455).

Lines 303-309: Please include specific values/numbers/ranges in the text for engraftment rates, percentage GFP+, and gene editing levels rather than using descriptive terms such as ‘substantial loss of gene editing levels’, ‘engrafted comparably’, ‘highest GFP expression’.

Response: Yes, values are now added to the results section where relevant (line 287-300).

Line 335-337: The authors state that “In contrast, using D10A Cas9n did not significantly lower CFU potential (Fig. 5e, Supplementary Fig. S14b), indicating very limited cytotoxicity of our D10A Cas9n gene editing strategy.” It would be more accurate to state that “In contrast, using D10A Cas9n did not significantly lower CFU frequency...”, since although D10A Cas9n did not significantly reduce the number of CFUs (Fig. 5e), it did appear to have significantly reduced the total number of cells per

plate in the CFU assay compared to mock-treated (Supplementary Fig. S14b; *p<0.05 for D10A Cas9n versus mock-treated), suggesting that CFU sizes may have been reduced by D10A Cas9n treatment. This might be regarded as decreasing “CFU potential” in terms of the overall proliferative potential of the cells within the colonies, although not in terms of the frequency of colony-forming units present in the treated HSPC population.

Response: Yes, this is true, we agree that the reduced size of colonies could be considered an indication of decreased CFU potential. We have rephrased the paragraph accordingly (line 328-332) and included information related to the colony size after treatment with the D10A Cas9n strategy.

Reviewer #4:

I co-reviewed this manuscript with one of the reviewers who provided the listed reports.

Response: Thanks, we appreciate the input.

We again thank the reviewers for their constructive feedback and believe that all comments have helped us improve the manuscript further.

Sincerely,
Jacob Giehm Mikkelsen